# Contextual and Seasonal LSTMs for Time Series Anomaly Detection

**Lingpei Zhang**[1], **Qingming Li**[1,*] **Yong Yang**[1], **Jiahao Chen**[1], **Rui Zeng**[1],
**Chenyang Lyu**[2], **Shouling Ji**[1,3]

[1]Zhejiang University, [2]Huawei Technologies
[3]Zhejiang Key Laboratory of Decision Intelligence
{lingpz, liqm, yangyong2022, xaddwell}@zju.edu.cn,
ruizeng24@outlook.com, lyuchenyang1@huawei.com, sji@zju.edu.cn

## Abstract

*Univariate time series (UTS)*, where each timestamp records a single variable, serve as crucial indicators in web systems and cloud servers. Anomaly detection in *UTS* plays an essential role in both data mining and system reliability management. However, existing reconstruction-based and prediction-based methods struggle to capture certain subtle anomalies, particularly small point anomalies and slowly rising anomalies. To address these challenges, we propose a novel prediction-based framework named Contextual and Seasonal LSTMs (CS-LSTMs). CS-LSTMs are built upon a noise decomposition strategy and jointly leverage contextual dependencies and seasonal patterns, thereby strengthening the detection of subtle anomalies. By integrating both time-domain and frequency-domain representations, CS-LSTMs achieve more accurate modeling of periodic trends and anomaly localization. Extensive evaluations on public benchmark datasets demonstrate that CS-LSTMs consistently outperform state-of-the-art methods, highlighting their effectiveness and practical value in robust time series anomaly detection.

## 1 Introduction

In cloud services, IoT, and monitoring systems (Faloutsos et al., 2019; Wen et al., 2022; Dai et al., 2021; Deldari et al., 2021; Ganatra et al., 2023; Günnemann et al., 2014; Huang et al., 2022; Kamarthi et al., 2022; Tuli et al., 2022; Xu et al., 2018), data is often represented as *univariate time series (UTS)* (Faloutsos et al., 2019), where each timestamp corresponds to a single variable. Anomalies may arise when a website is subject to malicious attacks or experiences excessive server load. These anomalies are typically reflected in *UTS* derived from traffic and system load monitoring. Missed or delayed detection of anomalous events may lead to severe consequences (Ruff et al., 2021). As illustrated in Figure 1, anomalies in *UTS* are typically categorized into two types: point anomalies, which refer to abrupt deviations (either global or contextual), and segment anomalies, which reflect more subtle shifts in trends or periodic patterns (Lai et al., 2021).

Unsupervised approaches, such as reconstruction-based and prediction-based methods, have become the dominant paradigm. Both aim to model normal behavior and detect anomalies by comparing predicted or reconstructed values against actual observations. Reconstruction-based techniques (Wang et al., 2024; Kingma et al., 2013) attempt to map abnormal segments to a normal space. However, lacking ground truth for anomalies leads to poor reconstruction and degraded performance. These methods also treat time windows independently, ignoring temporal dependencies. Prediction-based method (Zhou et al., 2021; Xu et al., 2021; Kingma et al., 2013) shows strong results in *multivariate time series (MTS)* using Transformers (Vaswani et al., 2017) or LSTMs (Hochreiter & Schmidhuber, 1997) to model complex dependencies. However, in *UTS*, each timestamp contains only one value, which restricts the representation of temporal context and results in suboptimal performance of these methods on *UTS* tasks.

---

*Corresponding author.

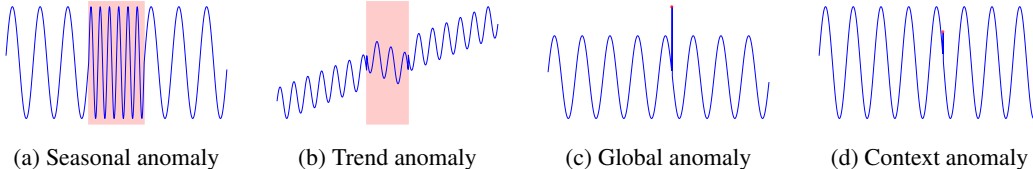

(a) Seasonal anomaly      (b) Trend anomaly      (c) Global anomaly      (d) Context anomaly

Figure 1: Anomaly types: (a), (b) segment anomalies; (c), (d) point anomalies. (a) Abrupt period shortening; (b) Unexpected trend drop; (c) Global outlier; (d) Contextual outlier within range but locally deviant.

To examine the limitations of existing anomaly detection methods in *UTS* tasks, we conduct a re-evaluation of several state-of-the-art approaches, including FCVAE (Wang et al., 2024), KAN-AD (Zhou et al., 2024), and TFAD (Zhang et al., 2022a). We find that two anomaly types remain largely undetected (Figure 2). (1) **Small Point Anomalies**: short-term spikes that appear normal over longer windows; (2) **Slowly Rising Anomalies**: gradual segment deviations from periodic patterns. Root cause analysis reveals that these hard-to-detect anomalies arise from existing methods' limitations. For example, FCVAE (Wang et al., 2024) focuses on frequency components rather than local dependencies, failing to capture subtle point anomalies or gradual segment anomalies; KAN-AD (Zhou et al., 2024) uses timing information for prediction but ignores frequency details.

To effectively detect the two types of anomalies, we highlight the following three key challenges:

**Challenge 1: Capturing Local Trends Rather Than Absolute Values**. As illustrated in Figure 3, the absolute magnitude of value changes is insufficient to determine whether a point is anomalous. For instance, in Figure 3a, changes between 0.5 and 1 are considered abnormal, whereas in Figure 3b, similar changes are deemed normal—only variations exceeding 5 are treated as anomalies. This observation underscores the importance of incorporating local information, i.e., data within a short time window, to predict future values. By modeling the value distribution and change patterns within the current local window, we can better estimate the expected range of future normal values.

**Challenge 2: Capturing the Evolution of Periodicity Rather Than Treating it as Static**. Time series data exhibit evolving trends and periodicity (Wen et al., 2021). Without future points, distinguishing slow anomalies from normal periodic changes is difficult. As shown in Figure 4, periodicity varies over time and between datasets, including period length and frequency changes. Previous methods focusing on adjacent historical periodicity neglect this evolution. Modeling dynamic yet heterogeneous periodicity is challenging. The Fourier transform (Cohen, 2020) effectively captures periodicity and its changes by converting time-domain data to the frequency domain.

**Challenge 3: Capturing Normal Patterns Amid Anomalies and Noise**. In real-world scenarios, time series often contain long-term noise and sudden anomalies, both of which hinder the learning of normal patterns. Noise introduces significant fluctuations, making it difficult for models to fit the underlying normal behavior accurately. Meanwhile, the lack of ground-truth labels for anomaly positions prevents the model from computing reliable prediction losses during training, further complicating the learning process.

To address these challenges, we propose CS-LSTMs, a novel prediction-based framework. CS-LSTMs are built upon a *noise decomposition strategy* to mitigate the impact of anomalies, and leverage both the local trends and the evolution of periodic patterns to predict the expected normal values at future time steps, and they detect anomalies via discrepancies between prediction and observation. CS-LSTMs employ a dual-branch architecture: one models long-term evolution in periodic, while the other captures short-term local fluctuations. By jointly modeling these complementary aspects, CS-LSTMs better capture time series data evolution and richer context, yielding more accurate and robust anomaly detection. Our contributions are summarized as follows:

- We analyze hard-to-detect anomalies in *UTS* and identify their root cause as the inability of existing methods to integrate local trends with periodic variations effectively.
- We introduce a noise decomposition strategy to better capture normal patterns by eliminating the non-stationary of time series effectively, thereby improving predictive accuracy and anomaly detection robustness.

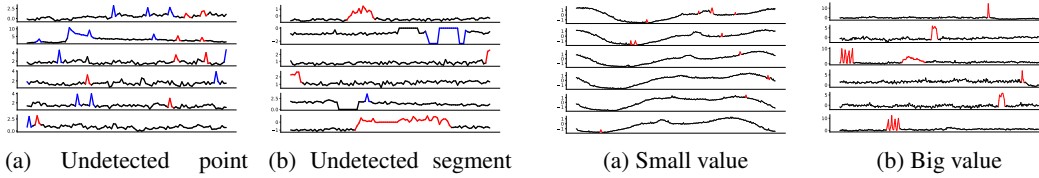

| (a) Undetected point anomaly | (b) Undetected segment anomaly | (a) Small value | (b) Big value |

Figure 2: Two anomaly types existing methods fail to detect. Normal points are shown in black, detected anomalies in blue, and undetected anomalies in red. (a) smaller point anomaly; (b) gradual rising segment anomaly.

Figure 3: Point anomalies at different scales. Normal points are black; anomalies are red. In (a), a value of 0.5 is anomalous due to the smooth background, while in (b), only a value around 5 is considered anomalous amid larger fluctuations. The key lies in context sensitivity.

- We design CS-LSTMs with a dual-branch architecture: one branch leverages frequency domain analysis (Hamilton, 2020) to model long-term periodicity and its evolution, while the other captures short-term local dynamics. The combination of the two effectively enhances the ability of anomaly detection.

- Extensive experiments demonstrate that CS-LSTMs achieve superior F1 scores and improve time efficiency by 40% over SOTA.

The replication package for this paper, including all our data, source code, and documentation, is publicly available online at https://github.com/NESA-Lab/Contextual-and-Seasonal-LSTMs-for-TSAD.

## 2 RELATED WORK

*Time Series Anomaly Detection (TSAD)* has been extensively studied in both univariate and multivariate settings. Existing approaches can be divided into traditional and deep learning methods, and further categorized into supervised and unsupervised depending on label availability.

Traditional methods (Lu & Ghorbani, 2008; Mahimkar et al., 2011; Rasheed et al., 2009; Rosner, 1983; Bailey, 1993; Vallis et al., 2014; Zhang et al., 2005; Guan et al., 2016; Lane et al., 1997; Peterson, 2009; Muthukrishnan et al., 2004) are computationally efficient and interpretable, making them suitable when data is scarce or temporal structures are clear. They include similarity-based approaches (e.g., kNN (Peterson, 2009)), window-based approaches using subsequence matching or HMMs (Gao et al., 2002), decomposition-based approaches (e.g., STL (Cleveland et al., 1990)), deviation-based methods such as SPOT (Vallis et al., 2014) based on Extreme Value Theory, and frequency-domain methods like FFT (Rasheed et al., 2009).

Deep learning approaches (Yang et al., 2023; Zhao et al., 2023b; Shen et al., 2020; Zong et al., 2018) have attracted increasing attention due to their strong representation learning capabilities and ability to capture complex temporal dependencies. Supervised methods (Laptev et al., 2015; Liu et al., 2015; Ren et al., 2019; Zhao et al., 2023b) achieve high accuracy with sufficient labels, often leveraging CNNs, random forests, or pseudo-labeling strategies. However, their generalization degrades under label scarcity or distribution shifts. Unsupervised methods (Chen et al., 2019; Kieu et al., 2022; Li et al., 2018; 2022b; Zhao et al., 2023a; Wang et al., 2024) are more common in practice, typically classified into reconstruction-based and prediction-based. Reconstruction-based approaches (e.g., FCVAE, Donut (Wang et al., 2024; Xu et al., 2018)) detect anomalies through reconstruction losses, while prediction-based methods forecast future values and flag deviations. Representative prediction models include LSTMAD and recent Transformer-based methods such as Informer (Zhou et al., 2021), Autoformer (Wu et al., 2021), and Fedformer (Zhou et al., 2022). More advanced architectures like Anomaly-Transformer (Xu et al., 2021) employ minimax strategies to enhance robustness, while TimesNet (Wu et al., 2022) integrates frequency-domain modeling for multi-scale and periodic signals. These approaches demonstrate strong adaptability in real-world scenarios with limited labeled data.

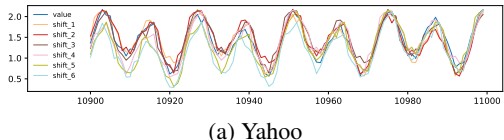 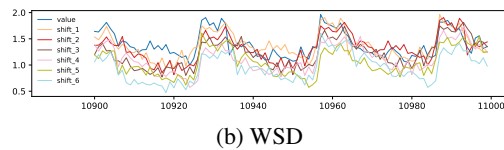

(a) Yahoo                                (b) WSD

Figure 4: Periodicity visualization in Yahoo and WSD. Time series are shifted and overlaid using different colors. Stronger overlap indicates stronger periodicity; reduced overlap over time suggests periodic changes.

## 3 PRELIMINARY

### 3.1 PROBLEM STATEMENT

This paper primarily addresses the issue of anomaly detection in single time series curves, also known as *UTS*. To elaborate on the problem more comprehensively, consider the following *UTS* observational data: $x_{0:t} = \{x_0, x_1, x_2, \ldots, x_t\}$ and anomaly labels $C = \{c_0, c_1, c_2, \ldots, c_t\}$, where $x_i \in \mathbb{R}, c_i \in \{0, 1\}, i \in \{0, 1, \ldots, t\}$, with $c = 0$ indicating a normal point and $c = 1$ indicating an anomaly. Given $x = [x_0, x_1, x_2, \ldots, x_t]$, the objective of *UTS* anomaly detection is to utilize the data $[x_0, x_1, \ldots, x_{i-1}]$ preceding each point $x_i$ to predict $c_i$.

### 3.2 LSTM AND COVARIATE

In our method, the LSTM is utilized to learn both the periodic trends and local variations of the time series data, while the value range information is incorporated as a covariate to assist in prediction. The combination of these two aspects effectively enhances the accuracy of the forecasting.

**LSTM** is a special type of Recurrent Neural Network (RNN) (Elman, 1990) designed to address the problem of vanishing gradient, which is difficult for traditional RNNs to capture long-term dependencies. LSTM is effective in handling time series data, natural language processing tasks, and other problems that involve temporal dependencies.

**Covariate** is a variable used as input in statistical models, machine learning, and associated research studies. It may be associated with the target variable (dependent variable) and could have a direct or indirect influence on it. Covariates (Hyndman & Khandakar, 2008) are typically used to explain variations in the target variable or to control confounding factors.

## 4 METHODOLOGY

Our method involves three main steps: **(1) Data Preprocessing**, where normalization and missing value imputation (Wen et al., 2020; Gao et al., 2020) are applied without relying on data augmentation (Le Guennec et al., 2016; Li et al., 2021); **(2) Dual-Branch Prediction Network**, where the S-LSTM branch learns periodic patterns from the frequency domain of the historical sequence, and the C-LSTM branch captures local trends and short-term variations; **(3) Anomaly Scoring with Masked Probabilistic Loss**, where we use a masked negative log-likelihood loss to focus on learning the distribution of normal data and predict future values. Anomalies are detected when the deviation between predicted and actual values exceeds the normal range.

### 4.1 NETWORK ARCHITECTURE

Our proposed *CS-LSTMs* model is demonstrated in Figure 5. The model consists of two branches: the first branch is the seasonal prediction branch, which predicts future points by learning the periodicity of historical sequences; and the second branch is the local prediction branch, which predicts future points by learning the changing trends and distributions in the short term.

### 4.1.1 NOISE DECOMPOSITION

The proposed noise decomposition strategy (Eq. 1) is designed to address two key challenges: (1) the absence of ground-truth labels for anomalous points; (2) the difficulty in fitting normal patterns of time series. In the context of time series forecasting, DLinear (Zeng et al., 2023) introduces a simple pooling-based decomposition method that splits the series into trend and residual components. In time series anomaly detection, TFAD (Zhang et al., 2022a) uses Robust STL (Wen et al., 2019) to decompose the series into trend and residual components for anomaly detection.

However, the objective of *CS-LSTMs* is to predict both periodic and trend components separately. Therefore, we propose a noise decomposition strategy that formulates time series as:

$$\text{Time Series} = (\text{Trend} + \text{Season}) + \text{Noise} \tag{1}$$

where only the noise is filtered out without further decomposition. DLinear's strategy is insufficient to address this formulation, while STL is overly complex and computationally expensive. To achieve efficient decomposition, we adopt a wavelet transform (Donoho & Johnstone, 1994):

$$\{c_A, c_D^{(L)}, \dots, c_D^{(1)}\} = \text{wavedec}(x, \psi, L) \tag{2}$$

$$\sigma_i = \frac{\text{median}(|c_D^{(i)}|)}{\Phi^{-1}(0.75)}, \quad \lambda_i = \sigma_i \sqrt{2\log n}, \quad \forall i = 1, \dots, L \tag{3}$$

$$\hat{c}_D^{(i)} = \text{sign}(c_D^{(i)}) \cdot \max(|c_D^{(i)}| - \lambda_i, 0), \quad \forall i = 1, \dots, L \tag{4}$$

$$\hat{x} = \text{waverec}\left(c_A, \hat{c}_D^{(L)}, \dots, \hat{c}_D^{(1)}\right) \tag{5}$$

We employ a method based on the *Median Absolute Deviation (MAD)* (Rousseeuw & Hubert, 2011). This approach is less sensitive to outliers compared to methods using the mean and standard deviation, allowing for a more accurate estimation of the data's dispersion. We assume that the underlying data (without outliers) follows a normal distribution $\Phi$.

The original signal $x \in \mathbb{R}^n$ is first decomposed into a set of wavelet coefficients using a selected wavelet basis $\psi$ and a predefined decomposition level $L$ (Eq. 2). Specifically, this decomposition yields one set of approximation coefficients at the coarsest level and multiple sets of detail coefficients at each level from 1 to $L$. The noise level $\sigma$ is then estimated based on the *MAD* of the finest-scale detail coefficients (Eq. 3), where the factor $\Phi^{-1}(0.75)$ ensures consistency with the standard deviation under the assumption of a Gaussian distribution.

Next, a universal threshold $\lambda$ is computed (Eq. 3), which is subsequently applied to all detail coefficients using soft-thresholding (Eq. 4). This process suppresses noise while preserving significant signal components. Finally, the denoised signal is reconstructed via the inverse wavelet transform (Eq. 5). The derivation of the denoising method is presented in the appendix D.

### 4.1.2 S-LSTM

The S-LSTM branch focuses on learning the periodic information and evolving trends in historical data to detect segment-level anomalies. As highlighted in Challenge 2, time series periodicity is dynamic rather than static. Unlike previous methods that only capture periodicity near the point of interest, S-LSTM models the temporal evolution of periodic patterns across the entire historical sequence, providing a more comprehensive representation of the underlying structure.

Specifically, we divide the historical sequence before the detection point into several equal-sized, non-overlapping windows. Each window is transformed from the time domain to the frequency domain via the Fourier transform, yielding frequency components that intuitively reflect the periodicity of the data. Since periodic characteristics vary across different segments, we model the relationships among these frequency representations to capture the evolution of periodicity over time.

To achieve this, a single-layer LSTM is employed to learn the variation trends of consecutive frequency vectors and predict future periodic patterns. Assuming a window size of $w$, the Fourier transform produces frequency vectors of a dimension $w_s$. Extracting $n$ such windows results in the input $z_s \in \mathbb{R}^{n \times w_s}$. Furthermore, to enhance the model's representation of temporal distribution,

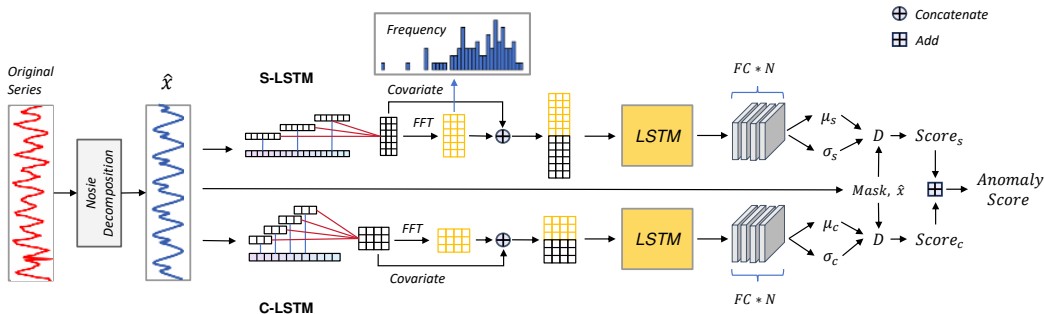

Figure 5: Architecture of CS-LSTMs.

the raw time-domain values are treated as covariates and concatenated with the frequency features before being fed into the LSTM. This combination enables the model to capture both distributional and periodic correlations effectively.

### 4.1.3 C-LSTM

The C-LSTM branch is designed to learn locally changing trends and distributions in the adjacent historical data to identify sudden, mutational anomalies. Due to the limited information available at individual points in *UTS*, conventional point-based prediction methods often struggle in anomaly detection tasks. Therefore, the C-LSTM branch captures richer local temporal dependencies by focusing on overlapping time segments.

Similarly, we extract a shorter historical sequence before the point of interest and divide it into several equal-sized, overlapping windows. This transformation shifts the task from learning relationships between individual points to learning relationships between segments, effectively alleviating the scarcity of information at single time points. Each overlapping window is further converted to the frequency domain via the Fourier transform to capture finer-grained details.

Next, a single-layer LSTM is employed to model the temporal relationships among these overlapping frequency segments, using this local information to predict future values. When a sudden change occurs, the predicted points will significantly deviate from the normal trend, enabling anomaly detection. Given the need to capture short-term local variations, we select a smaller window size $w_c$, intercepting $n$ overlapping windows to form the input $z_c \in \mathbb{R}^{n \times w_c}$ for the model, thus achieving precise modeling of local temporal dynamics.

### 4.2 Loss Function

We adopt a noise-decomposed *Negative Log-Likelihood (NLL)* loss function to perform anomaly detection by modeling the probabilistic distribution of normal points. This approach emphasizes filtering anomalous regions and accurately capturing normal patterns. Since normal behavior is inherently probabilistic rather than deterministic, a key challenge lies in how to represent its variation range appropriately.

To address this, we design a loss function with three core components: (1) noise decomposition, (2) anomaly masking, and (3) range-based prediction. During loss calculation, we leverage anomaly masks and noise decomposition results to obtain the reference value for each point under normal conditions. This enables the model to focus on learning normal patterns without being misled by unreliable ground truth in anomalous regions.

Compared with *MSE* (Thompson, 1990) or *MAE* (Hodson, 2022), the NLL loss better reflects uncertainty by predicting both mean and variance, naturally adapting to the probabilistic characteristics of normal behavior.

The final loss function $\mathcal{D}$ (Eq. 6) incorporates the predicted mean $\mu$, variance $\sigma^2$, ground truth $x$, noise-decomposed value $\hat{x}$, and the anomaly mask. By optimizing only on reliable data, our model achieves more robust normal pattern learning. We provide the proof of the loss function's

Table 1: The performance of anomaly detection on the test datasets. F1 means best F1 and F1* means delay F1. The best-performing results are shown in bold, while the second-best results are underlined.

| Method | Yahoo | | | | | | KPI | | | | | | WSD | | | | | | NAB | | | | | |
|---|---|---|---|---|---|---|---|---|---|---|---|---|---|---|---|---|---|---|---|---|---|---|---|---|
| | F1 | P | R | F1* | P | R | F1 | P | R | F1* | P | R | F1 | P | R | F1* | P | R | F1 | P | R | F1* | P | R |
| SPOT | 0.417 | 0.572 | 0.328 | 0.417 | 0.572 | 0.328 | 0.360 | 0.966 | 0.221 | 0.143 | 0.911 | 0.077 | 0.472 | 0.947 | 0.315 | 0.237 | 0.887 | 0.137 | 0.829 | 0.992 | 0.713 | 0.829 | 0.992 | 0.712 |
| SRCNN | 0.251 | 0.268 | 0.236 | 0.198 | 0.219 | 0.181 | 0.786 | 0.673 | 0.944 | 0.678 | 0.617 | 0.753 | 0.170 | 0.093 | 0.903 | 0.053 | 0.028 | 0.361 | 0.828 | 0.825 | 0.832 | 0.575 | 0.460 | 0.766 |
| DONUT | 0.215 | 0.381 | 0.150 | 0.215 | 0.381 | 0.150 | 0.454 | 0.378 | 0.569 | 0.364 | 0.328 | 0.407 | 0.224 | 0.263 | 0.195 | 0.158 | 0.199 | 0.131 | 0.935 | 0.933 | 0.937 | 0.797 | 0.821 | 0.774 |
| VQRAE | 0.510 | 0.706 | 0.399 | 0.492 | 0.691 | 0.381 | 0.272 | 0.202 | 0.418 | 0.137 | 0.167 | 0.117 | 0.312 | 0.518 | 0.233 | 0.103 | 0.241 | 0.066 | 0.933 | 0.990 | 0.882 | 0.893 | 0.806 | 1.000 |
| Anotransfer | 0.567 | 0.902 | 0.413 | 0.496 | 0.575 | 0.437 | 0.685 | 0.815 | 0.591 | 0.461 | 0.557 | 0.394 | 0.674 | 0.695 | 0.654 | 0.379 | 0.331 | 0.444 | 0.965 | 0.962 | 0.968 | 0.871 | 0.837 | 0.908 |
| Informer | 0.694 | 0.747 | 0.648 | 0.671 | 0.731 | 0.619 | 0.918 | 0.927 | 0.910 | 0.822 | 0.801 | 0.845 | 0.557 | 0.532 | 0.583 | 0.393 | 0.402 | 0.385 | 0.973 | 0.971 | 0.974 | 0.892 | 0.878 | 0.907 |
| TFAD | 0.792 | 0.875 | 0.723 | 0.791 | 0.879 | 0.719 | 0.752 | 0.684 | 0.834 | 0.680 | 0.650 | 0.714 | 0.628 | 0.541 | 0.750 | 0.455 | 0.431 | 0.482 | 0.734 | 0.749 | 0.719 | 0.248 | 0.265 | 0.233 |
| Anomaly-Transformer | 0.274 | 0.588 | 0.179 | 0.029 | 0.054 | 0.020 | 0.868 | 0.930 | 0.814 | 0.346 | 0.622 | 0.240 | 0.728 | 0.861 | 0.630 | 0.137 | 0.144 | 0.129 | 0.971 | 0.944 | 1.000 | 0.911 | 0.891 | 0.932 |
| FCVAE | 0.854 | 0.913 | 0.800 | 0.839 | 0.911 | 0.778 | 0.924 | 0.929 | 0.919 | 0.851 | 0.894 | 0.812 | 0.805 | 0.757 | 0.850 | 0.696 | 0.562 | 0.913 | 0.972 | 0.946 | 1.000 | 0.899 | 0.888 | 0.910 |
| KAN-AD | 0.679 | 0.634 | 0.726 | 0.668 | 0.630 | 0.711 | 0.912 | 0.919 | 0.905 | 0.760 | 0.744 | 0.777 | 0.756 | 0.881 | 0.663 | 0.680 | 0.840 | 0.572 | 0.990 | 0.981 | 1.000 | 0.900 | 0.890 | 0.911 |
| CS-LSTMs | **0.885** | 0.919 | 0.853 | **0.878** | 0.915 | 0.845 | **0.936** | 0.918 | 0.955 | **0.879** | 0.905 | 0.855 | **0.910** | 0.898 | 0.921 | **0.857** | 0.891 | 0.825 | **0.996** | 0.992 | 1.000 | **0.918** | 0.936 | 0.901 |

effectiveness and convergence in the appendix F.

$$\mathcal{D}(\mu, \sigma, x, \hat{x}) = \log \sigma^2 + \frac{(x \odot \text{mask} + \hat{x} \odot \tilde{\text{mask}} - \mu)^2}{\sigma^2} \quad (6)$$

### 4.2.1 THE OVERALL FRAMEWORK

Both branches predict future values from historical inputs. Let $\mu_s, \sigma_s = \text{S-LSTM}(z_s)$ and $\mu_c, \sigma_c = \text{C-LSTM}(z_c)$ denote the predicted values from each branch, respectively. Their respective losses, $L_s$ and $L_c$, are computed based on the discrepancy between predicted values and actual future points, respectively. The total loss is designed as follows:

$$L_s = \mathcal{D}(\mu_s, \sigma_s, x, \hat{x}), \quad L_c = \mathcal{D}(\mu_c, \sigma_c, x, \hat{x}), \quad L = L_s + L_c \quad (7)$$

As shown in Figure 8, the S-LSTM branch segments the time series into windows of size $N \times w_s$, transforms them into the frequency domain via the Fourier Transform, and predicts the next cycle. The result is converted back to the time domain using the Inverse Fourier Transform to produce $\mu_s$ and $\sigma_s$. As shown in Figure 9, in the C-LSTM branch, the series is divided into overlapping windows of size $N \times w_c$, with $w_c$ slightly larger than the stride to retain local patterns. These are concatenated into $z_c$ and processed by C-LSTM to yield $\mu_c$ and $\sigma_c$.

## 5 EXPERIMENTS

### 5.1 EXPERIMENTS SETTINGS

#### 5.1.1 DATASETS

To evaluate the effectiveness of our proposed framework, we conducted experiments on four benchmark datasets. The Yahoo dataset (n. d., b) is an open dataset specifically designed for anomaly detection, released by Yahoo Labs. The KPI dataset (Li et al., 2022a) consists of data collected from five major internet companies—Sougo, eBay, Baidu, Tencent, and Ali—capturing key performance indicators (KPI). The WSD dataset (n. d., a) contains real-world KPIs collected from three prominent Internet companies: Baidu, Sogou, and eBay, representing large-scale web service usage. Finally, the NAB dataset (Lavin & Ahmad, 2015), created by Numenta, is an open benchmark designed to evaluate the performance of *TSAD* algorithms. For all publicly available datasets, we divide them into 35% training set, 15% validation set, and 50% testing set.

Table 2: Dataset statistics. "Total Length" indicates the total number of time series data. "Anomaly Ratio" represents the proportion of anomaly points in the entire time series data.

| Datasets | Total Length | Anomaly Ratio |
|---|---|---|
| Yahoo | 573K | 0.68% |
| KPI | 5923K | 2.26% |
| WSD | 7511K | 1.60% |
| NAB | 159K | 9.89% |

#### 5.1.2 BASELINES

We compare CS-LSTMs against ten representative methods: SPOT (Siffer et al., 2017), SR-CNN (Ren et al., 2019), TFAD (Zhang et al., 2022a), DONUT (Xu et al., 2018), Informer (Zhou et al., 2021), Anomaly-Transformer (Xu et al., 2021), AnoTransfer (Zhang et al., 2022b),

Table 3: Comparison of transferability among five mainstream models. The left side reports results when trained on Yahoo and tested on KPI, WSD, and NAB; the right side shows results when trained on KPI and tested on Yahoo, WSD, and NAB. **F1** denotes the best F1 score, **P** denotes Precision, and **R** denotes Recall.

| Method | Yahoo | | | | | | | | | KPI | | | | | | | | |
| | KPI | | | WSD | | | NAB | | | Yahoo | | | WSD | | | NAB | | |
| | F1 | P | R | F1 | P | R | F1 | P | R | F1 | P | R | F1 | P | R | F1 | P | R |
| Informer | 0.901 | 0.895 | 0.907 | 0.571 | 0.552 | 0.591 | 0.969 | 0.970 | 0.971 | 0.536 | 0.552 | 0.521 | 0.564 | 0.545 | 0.584 | 0.971 | 0.964 | 0.978 |
| TFAD | 0.810 | 0.815 | 0.805 | 0.615 | 0.554 | 0.690 | 0.714 | 0.729 | 0.699 | 0.594 | 0.864 | 0.453 | 0.635 | 0.570 | 0.717 | 0.727 | 0.751 | 0.706 |
| Anomaly-Transformer | 0.791 | 0.701 | 0.906 | 0.746 | 0.616 | 0.945 | 0.957 | 0.917 | 1.000 | 0.403 | 0.376 | 0.435 | 0.731 | 0.598 | 0.940 | 0.957 | 0.918 | 1.000 |
| FCVAE | 0.915 | 0.914 | 0.916 | 0.829 | 0.916 | 0.759 | 0.968 | 0.938 | 1.000 | 0.574 | 0.858 | 0.431 | 0.859 | 0.833 | 0.886 | 0.967 | 0.935 | 1.000 |
| KAN-AD | 0.905 | 0.906 | 0.903 | 0.743 | 0.883 | 0.641 | **0.991** | 0.982 | 1.000 | 0.507 | 0.439 | 0.601 | 0.754 | 0.893 | 0.652 | **0.990** | 0.981 | 1.000 |
| CS-LSTMs | **0.929** | 0.890 | 0.971 | **0.883** | 0.920 | 0.848 | 0.986 | 0.972 | 1.000 | **0.670** | 0.794 | 0.579 | **0.897** | 0.927 | 0.868 | **0.990** | 0.980 | 1.000 |

VQRAE (Kieu et al., 2022), KAN-AD (Zhou et al., 2024) and FCVAE (Wang et al., 2024). SPOT is a statistical method based on extreme value theory. SRCNN and TFAD are supervised methods requiring high-quality labels. DONUT, VQRAE, and AnoTransfer are unsupervised reconstruction-based VAEs. Informer is a prediction-based model using attention, while Anomaly-Transformer applies Transformers for unsupervised outlier detection. FCVAE leverages frequency-conditioned VAEs for sequence reconstruction, treating all points in a window as normal.

### 5.1.3 EVALUATION METRICS

In industrial scenarios, anomalies often appear as segments rather than isolated points, making point-wise precision less critical. Following prior work (Xu et al., 2018; Ren et al., 2019), we use two metrics: **Best F1** and **Delay F1**, both employing point adjustment strategies. Best F1 counts a segment as detected if any point within it is identified, while Delay F1 requires detection within a maximum waiting time $k$, with smaller $k$ enforcing stricter evaluation.

Figure 6: Illustration of the adjustment strategy.

We set dataset-specific $k$ values based on complexity and noise: Yahoo ($k = 3$) and KPI ($k = 7$) have accurate annotations; NAB ($k = 150$) contains many continuous segment anomalies; WSD ($k = 40$) has substantial label noise around anomalies, requiring a moderate increase in $k$ to reduce impact.

### 5.2 EFFECTIVENESS AND EFFICIENCY

Table 1 compares CS-LSTMs with nine baselines. Our model consistently outperforms all competitors, achieving Best F1 gains of 3.1%, 1.2%, 10.5%, and 0.6%, and Delay F1 gains of 3.9%, 2.8%, 16.1%, and 0.7% across four datasets.

Shown in Table 4, CS-LSTMs are also efficient, with 600K parameters—less than half of SOTA's 1.4M—reducing inference time by 40%, enabling faster anomaly alerts. Baseline limitations include SPOT (Siffer et al., 2017) misclassifying mild anomalies, SRCNN (Ren et al., 2019) struggling with implicit features, Informer (Zhou et al., 2021) missing frequency changes, Anomaly-Transformer (Xu et al., 2021) showing lower Delay F1, TFAD (Zhang et al., 2022a) suffering latency, and FCVAE (Wang et al., 2024) failing on subtle spikes or gradual periodic shifts.

Table 4: Inference efficiency of 512 iterations.

| Method | GPU 3090 24G | | |
| | GPU Memory | GPU Time | CPU Time |
| Informer | 147 MB | 2280 ms | 2280 ms |
| TFAD | 130 MB | 40 ms | 40 ms |
| Ano-Transformer | 130 MB | 1539 ms | 1539 ms |
| FCVAE | 37 MB | 7.73 ms | 7.73 ms |
| CS-LSTMs | 44 MB | 4.62 ms | 3.82 ms |

By jointly modeling time and frequency domain information and forecasting future normal points, CS-LSTMs more effectively detect subtle abrupt changes and smooth periodic variations.

### 5.3 TRANSFERABILITY EXPERIMENTS

In industrial scenarios, retraining *TSAD* models for each new domain is costly and often restricted by privacy concerns, making cross-domain transferability essential. Considering that WSD and NAB contain significant label noise, we select Yahoo and KPI as source datasets to train models on these datasets, and directly test them on the remaining three to evaluate their robustness under varying anomaly types and noise levels.

As shown in Table 3, we selected the best performing model in Table 1 for the transferability experiment. Baseline models including Informer (Zhou et al., 2021), TFAD (Zhang et al., 2022a), Anomaly-Transformer (Xu et al., 2021), and FCVAE (Wang et al., 2024), suffer significant performance drops on unseen datasets, while CS-LSTMs consistently achieve the best F1 scores across all transfer tasks. This advantage stems from its dual-branch design: S-LSTM captures global periodic patterns in the frequency domain, while C-LSTM models local trend variations via overlapping windows. Combined with unsupervised and noise-resilient training, CS-LSTMs demonstrate strong cross-domain adaptability and practical deployment potential.

### 5.4 ABLATION STUDY

To validate our model design, we conduct ablation studies on three key components and the window size. Results show that each component significantly contributes to improved performance in time series prediction and anomaly detection. Meanwhile, the window size also has a certain degree of influence on the model's capability. The appropriate sizes of the seasonal window and the contextual window can improve the model's performance.

#### 5.4.1 KEY COMPONENTS

We conducted ablation studies on different components of CS-LSTMs, including the co-branch architecture, covariate, and the noise decomposition module. Regarding the architecture, we evaluated the S-LSTM branch for modeling periodic patterns and the C-LSTM branch for capturing local dynamics. Using either branch alone yielded inferior results compared to the combined model, indicating their complementarity. Meanwhile, the results show that incorporating covariates leads to better performance than excluding them, since frequency-domain analysis can capture periodicity, while temporal information provides complementary local variation features. For the noise decomposition module, the results demonstrate that this design facilitates the modeling of normal patterns and enhances anomaly detection performance. Overall, the results in Table 5 confirm that the dual-branch architecture, the introduction of covariates, and the noise decomposition strategy are all indispensable for achieving superior performance.

Table 5: Ablation experiments across Yahoo, KPI, and WSD datasets.

| Method | Yahoo | | KPI | | WSD | | NAB | |
|---|---|---|---|---|---|---|---|---|
| | Best | Delay | Best | Delay | Best | Delay | Best | Delay |
| w/o C-LSTM | 0.864 | 0.862 | 0.923 | 0.849 | 0.856 | 0.725 | 0.985 | 0.905 |
| w/o S-LSTM | 0.717 | 0.768 | 0.904 | 0.809 | 0.762 | 0.431 | 0.987 | 0.902 |
| w/o Covariate | 0.826 | 0.816 | 0.925 | 0.850 | 0.840 | 0.576 | 0.986 | 0.901 |
| w/o Noise Decomposition and Mask | 0.868 | 0.858 | 0.913 | 0.864 | 0.858 | 0.812 | 0.972 | 0.897 |
| CS-LSTMs | **0.885** | **0.878** | **0.936** | **0.879** | **0.910** | **0.857** | **0.996** | **0.918** |

#### 5.4.2 DENOISE STRATEGY

To evaluate the impact of different denoising strategies, we conducted a set of ablation studies. Specifically, we tested two alternative approaches, Pooling Decomposition and STL Decomposition, as substitutes for our proposed denoising module, and compared all three strategies across four benchmark datasets. As shown in Table 6, our method consistently achieves the best anomaly-detection performance on all datasets. Table 7 further reports the per-epoch training time under each strategy, where our approach again demonstrates superior efficiency.

Upon analysis, we find that Pooling Decomposition is overly coarse-grained: although fast, it struggles to capture fine-level temporal structure, leading to suboptimal accuracy. In contrast, STL

Table 6: Comparison of denoising strategies across Yahoo, NAB, WSD, and AIOPS datasets.

| Method | Yahoo | | NAB | | WSD | | AIOPS | |
|---|---|---|---|---|---|---|---|---|
| | Best | Delay | Best | Delay | Best | Delay | Best | Delay |
| Noise Decomposition | **0.885** | **0.878** | **0.996** | **0.918** | **0.910** | **0.857** | **0.936** | **0.879** |
| Pooling Decomposition | 0.872 | 0.860 | 0.988 | 0.904 | 0.865 | 0.812 | 0.923 | 0.871 |
| STL Decomposition | 0.877 | 0.871 | 0.989 | 0.890 | 0.874 | 0.816 | 0.914 | 0.875 |

Decomposition offers more fine-grained modeling but incurs substantial computational overhead. Overall, our denoising module strikes a more favorable balance between accuracy and efficiency, achieving the best performance among all compared methods.

Table 7: Training time per epoch under different denoising strategies.

| Method | Time |
|---|---|
| Noise Decomposition | **52.383s** |
| Pooling Decomposition | 59.332s |
| STL Decomposition | 458.389s |

### 5.4.3 WINDOW SIZE

We also study how window sizes affect CS-LSTMs' performance across four datasets (Yahoo, KPI, WSD, NAB), as shown in Figure 7. The seasonal branch (*S-LSTM*) benefits from windows spanning a full period, improving frequency-domain modeling, but degrades when longer windows introduce noise. The context branch (*C-LSTM*) improves with larger windows until an optimal point, after which fine-grained changes are diluted. Overall, CS-LSTMs remain robust across window sizes, achieving strong F1 scores. In practice, periodicity guides seasonal window selection, while context windows show good cross-dataset generalizability, reducing manual tuning.

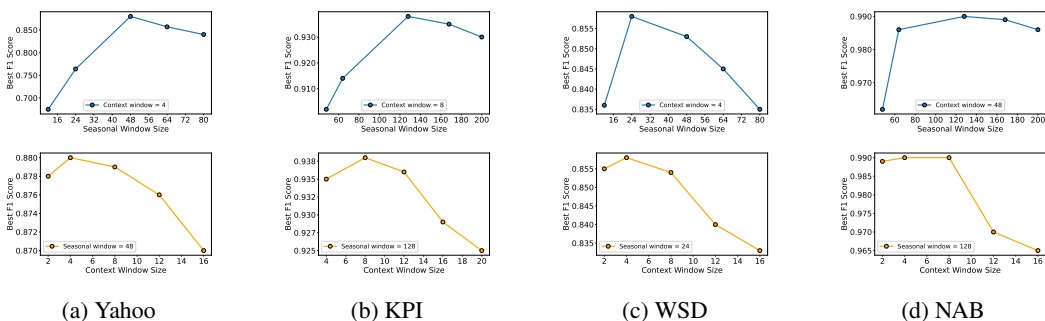

| (a) Yahoo | (b) KPI | (c) WSD | (d) NAB |
|---|---|---|---|

Figure 7: Impact of window sizes on F1 score. Top: Varying context window (blue) with fixed seasonal window. Bottom: Varying seasonal window (orange) with fixed context window.

## 6 CONCLUSION

This paper presents CS-LSTMs, a novel unsupervised prediction-based method for anomaly detection in *UTS*. The model adopts a dual-branch architecture: S-LSTM captures long-term periodic patterns via frequency-domain modeling with FFT, while C-LSTM focuses on short-term local trends through context dependencies. The combination of the two branches improves the capabilities of the model. We introduce noise decomposition during training to reduce noise from abnormal points and propose a "normal distance" metric to quantify anomaly severity. CS-LSTMs address key challenges in *UTS* anomaly detection by effectively modeling both periodic structures and local variations. It achieves SOTA results on four public datasets (Yahoo, KPI, WSD, NAB), and improves inference efficiency by 40%.

## ACKNOWLEDGEMENTS

This work was partly supported by New Generation Artificial Intelligence-National Science and Technology Major Project (2025ZD0123503), NSFC under No. U2441239, U24A20336, 62172243, 62402425, 62402418, 62502432 and 62502433, the China Postdoctoral Science Foundation under No. 2024M762829 and 2025M781522, the Zhejiang Provincial Natural Science Foundation under No. LD24F020002, the "Pioneer and Leading Goose" R&D Program of Zhejiang under No. 2025C02033 and 2025C01082.

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

## APPENDIX

## A    HYPERPARAMETER SETTINGS

To ensure reproducibility, Table 8 summarizes the hyperparameter settings used in all experiments. CS-LSTMs adopt dataset-specific hyperparameters mainly due to differences in intrinsic information density (e.g., Yahoo: 1-hour sampling, NAB: 5-minute sampling, WSD/AIOps: 1-minute sampling). Similar practices are followed in FCVAE (Wang et al., 2024) and TFAD (Zhang et al., 2022a).

In practice, only three hyperparameters need to be set: **total_window_size** (input sequence length), **seasonal_window_size** (S-LSTM window), and **contextual_window_size** (C-LSTM window). The first and last can be derived from **seasonal_window_size** using simple overlapping windows, without requiring extensive tuning. Practical rules are:

- To capture periodicity evolution:

$$\texttt{total\_window\_size} = m \times \texttt{seasonal\_window\_size}, \quad m \in (5, 7)$$

- To capture contextual variations:

$$\texttt{contextual\_window\_size} = \texttt{seasonal\_window\_size}//n, \quad n \in (5, 7)$$

These settings consistently yield stable performance across datasets. As shown in Figure 7, performance varies by only about 2% within a reasonable range of window sizes, indicating robustness and ease of deployment.

Table 8: Hyperparameter settings used in all experiments.

| Hyperparameter | Value |
|---|---|
| batch_size | 512 |
| max_epochs | 30 |
| seasonal_window_size | 48 |
| total_window_size | 240 |
| context_window_size | 4 |
| d_model | 256 |
| num_layers | 1 |

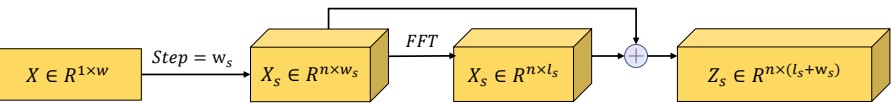

Figure 8: Architecture of S-LSTM.

## B ARCHITECTURE OF C-LSTM AND S-LSTM

## C LSTM

LSTM is a special type of Recurrent Neural Network (RNN) (Elman, 1990) designed to address the problem of vanishing gradient, which is difficult for traditional RNNs to capture long-term dependencies. LSTM is effective in handling time series data, natural language processing tasks, and other problems that involve temporal dependencies. The key components of LSTM are the *Memory Cell* and *Gating Mechanism*, which allow the network to selectively retain or discard information.

**Forget Gate:** The forget gate determines which information in the memory cell should be discarded. The mathematical expression is:

$$f_t = \sigma(W_f \cdot [h_{t-1}, x_t] + b_f) \tag{8}$$

where $f_t$ is the output of the forget gate, and $\sigma$ represents the sigmoid activation function.

**Input Gate:** The input gate decides what new information should be added to the memory cell:

$$i_t = \sigma(W_i \cdot [h_{t-1}, x_t] + b_i) \tag{9}$$

The input gate is combined with the candidate memory cell:

$$\tilde{C}_t = \tanh(w_s \cdot [h_{t-1}, x_t] + b_c) \tag{10}$$

**Memory Update:** The memory cell is updated by combining the forget gate and input gate:

$$C_t = f_t \cdot C_{t-1} + i_t \cdot \tilde{C}_t \tag{11}$$

**Output Gate:** The output gate controls the information output at the current time step:

$$o_t = \sigma(W_o \cdot [h_{t-1}, x_t] + b_o) \tag{12}$$

The final output is:

$$h_t = o_t \cdot \tanh(C_t) \tag{13}$$

## D NOISE DECOMPOSING WITH MAD-BASED THRESHOLDING

To mitigate noise in one-dimensional time series, we adopt wavelet shrinkage with thresholds estimated via the Median Absolute Deviation (MAD). MAD uses the median instead of the mean, minimizing the impact of extreme values on dispersion. Let $x = \{x_t\}_{t=1}^N$ denote the observed signal, which is assumed to be corrupted by additive white Gaussian noise. The signal is first decomposed into approximation and detail coefficients using a discrete wavelet transform (DWT):

$$x(t) \xrightarrow{\text{DWT}} c_A^{(j)}, \{c_D^{(j)}\}_{j=1}^L$$

where $c_A^{(j)}$ are approximation coefficients at level $j$, $c_D^{(j)}$ are detail coefficients, and $L$ is the decomposition level. High-frequency detail coefficients $c_D^{(L)}$ predominantly capture noise.

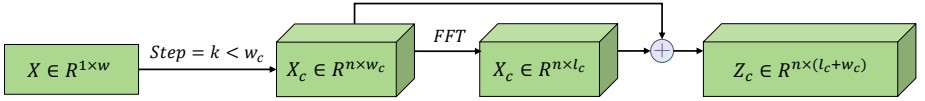

Figure 9: Architecture of C-LSTM.

**Noise estimation via MAD.** Generally, we assume that the noise follows a normal distribution, the noise standard deviation $\sigma_i$ is estimated from the detail coefficients $c_D^{(i)}$ by

$$\sigma_i = \frac{\text{median}\left(|c_D^{(i)}|\right)}{\Phi^{-1}(0.75)}, \quad \forall i = 1, \ldots, L \tag{14}$$

The Median Absolute Deviation (MAD) is defined as

$$\text{MAD}(x) = \text{median}\left(|x_i - \text{median}(x)|\right)$$

and serves as a robust alternative to the standard deviation. To calibrate MAD under Gaussian noise, consider $Z \sim \mathcal{N}(0, 1)$. Then

$$\text{MAD}(Z) = \text{median}(|Z|)$$

Let $Y = |Z|$. Its distribution satisfies

$$P(Y \le y) = P(-y \le Z \le y) = 2\Phi(y) - 1$$

where $\Phi(\cdot)$ denotes the standard normal cumulative distribution function. The median $m$ of $Y$ solves

$$2\Phi(m) - 1 = 0.5 \quad \Rightarrow \quad \Phi(m) = 0.75$$

yielding

$$m = \Phi^{-1}(0.75) \approx 0.6745$$

Thus, for Gaussian noise with variance $\sigma^2$

$$\text{MAD} \approx 0.6745\sigma$$

To obtain a consistent estimator of the noise standard deviation, MAD is normalized as

$$\sigma \approx \frac{\text{MAD}}{\Phi^{-1}(0.75)}$$

Hence, the constant $\Phi^{-1}(0.75)$ arises as the $0.75$-quantile of the standard normal distribution, ensuring that MAD yields an asymptotically unbiased estimate of $\sigma$ under Gaussian assumptions.

**Universal threshold.** Following VisuShrink, the universal threshold (Donoho & Johnstone, 1994) is defined as

$$\lambda_i = \sigma_i \sqrt{2 \log n}, \quad \forall i = 1, \ldots, L \tag{15}$$

where $N$ is the signal length. This threshold is then applied to each set of detail coefficients.

We consider wavelet coefficients $\{z_j\}_{j=1}^n$ generated purely by Gaussian noise, i.i.d. $z_j \sim \mathcal{N}(0, \sigma^2)$, and define the maximum absolute coefficient as $M_n := \max_{1 \le j \le n} |z_j|$. To control the probability that any noise coefficient exceeds a threshold $t > 0$, we apply the union bound:

$$\Pr(M_n > t) \le \sum_{j=1}^n \Pr(|z_j| > t) = n \Pr(|Z| > t/\sigma), \quad Z \sim \mathcal{N}(0, 1)$$

and invoke the standard Gaussian tail bound $\Pr(|Z| > u) \le \sqrt{\frac{2}{\pi}} \frac{e^{-u^2/2}}{u}$, yielding

$$\Pr(M_n > t) \le n \sqrt{\frac{2}{\pi}} \frac{\sigma}{t} \exp\left(-\frac{t^2}{2\sigma^2}\right)$$

Setting $t = \sigma\sqrt{2\log n}$ leads to

$$\Pr\left(M_n > \sigma\sqrt{2\log n}\right) \leq \sqrt{\frac{1}{\pi\log n}} \xrightarrow[n\to\infty]{} 0$$

which implies that, with high probability, all noise coefficients remain below $\sigma\sqrt{2\log n}$. Equivalently, a threshold of this order asymptotically removes coefficients generated solely by noise. From the perspective of extreme-value theory, the maximum of $n$ i.i.d. Gaussian samples satisfies

$$M_n = \sigma\sqrt{2\log n} + O\left(\frac{\log\log n}{\sqrt{\log n}}\right)$$

in probability, and, after centering and scaling, converges in distribution to the Gumbel law. The leading term $\sqrt{2\log n}$ thus captures the characteristic growth of the maximum and justifies the scaling of the universal threshold.

Practical implications follow directly: coefficients exceeding $\lambda = \sigma\sqrt{2\log n}$ are unlikely to originate from noise, while smaller coefficients are predominantly noise and can be suppressed. The universal threshold is therefore theoretically grounded, yet conservative; it guarantees noise removal but may introduce over-smoothing by discarding small yet informative coefficients. Adaptive alternatives, including SURE-based or level-dependent thresholds, can mitigate this bias. Refinements such as $\sigma\sqrt{2\log n - \log\log n}$ further improve finite-sample accuracy, while level-dependent thresholds $\lambda_i = \sigma_i\sqrt{2\log n}$ accommodate different variances across wavelet scales. Overall, the universal threshold arises from controlling the maximal deviation of Gaussian noise coefficients, providing a principled and asymptotically optimal rule to discriminate between noise and signal in the wavelet domain.

**Thresholding operator.** For a coefficient $c$, the soft-thresholding operator is given by

$$\hat{c}_D^{(i)} = \text{sign}(c_D^{(i)}) \cdot \max\left(|c_D^{(i)}| - \lambda_i, 0\right), \quad \forall i = 1, \ldots, L \tag{16}$$

which shrinks small coefficients towards zero while preserving larger structures. Hard-thresholding can also be applied, though it may introduce discontinuities.

**Reconstruction.** The denoised signal is reconstructed via the inverse wavelet transform (IDWT):

$$\hat{x}(t) = \text{IDWT}\left(c_A^{(j)}, \{\hat{c}_D^{(j)}\}_{j=1}^L\right)$$

**Implementation.** In practice, this procedure can be implemented using the Algorithm 1

---

**Algorithm 1:** Noise decomposing with MAD-based Thresholding

---

**Input:** Noisy signal $x \in \mathbb{R}^n$, wavelet basis $\psi$, decomposition level $L$
**Output:** Denoised signal $\hat{x}$
$\{c_A, c_D^{(L)}, \ldots, c_D^{(1)}\} \leftarrow \text{wavedec}(x, \psi, L)$
**for** $i \leftarrow 1$ **to** $L$ **do**

$\quad \sigma_i \leftarrow \frac{\text{median}(|c_D^{(i)}|)}{\Phi^{-1}(0.75)}$ ; /* MAD noise estimate */

$\quad \lambda_i \leftarrow \sigma_i \cdot \sqrt{2\log n}$ ; /* Universal threshold */

$\quad \hat{c}_D^{(i)} \leftarrow \text{sign}(c_D^{(i)}) \cdot \max(|c_D^{(i)}| - \lambda_i, 0)$ ; /* Soft-thresholding */

$\hat{x} \leftarrow \text{waverec}(c_A, \hat{c}_D^{(L)}, \ldots, \hat{c}_D^{(1)})$
**return** $\hat{x}$

---

This approach effectively suppresses high-frequency noise while preserving salient temporal structures, making it well-suited for anomaly detection and other downstream time series tasks.

# E   EVALUATING CS-LSTMS UNDER KEY CHALLENGES

To further demonstrate the effectiveness of CS-LSTMs, we conduct a case study focusing on the two key challenges previously discussed. Specifically, we evaluate the model's performance in two representative scenarios: (1) detecting **Small Point Anomalies**, which relates to the challenge of **Capturing Local Trends**, and (2) detecting **Slowly Rising Anomalies**, which reflects the challenge of **Capturing Periodicity and Its Evolution**.

As shown in Figure 10, we compare the detection results of CS-LSTMs with those of a state-of-the-art baseline, FCVAE (Wang et al., 2024). In this figure, black lines represent normal time points, blue lines indicate correctly detected anomalies, and red lines mark missed anomalies. FCVAE (Wang et al., 2024) struggles to simultaneously detect both point anomalies and segment anomalies within the same time series data. In contrast, CS-LSTMs achieve precise detection in both scenarios, successfully identifying all anomalous points and segments.

This superior performance is attributed to the model's dual-branch design: the C-LSTM branch effectively captures localized patterns and short-term variations, enabling accurate detection of mutational anomalies; meanwhile, the S-LSTM branch models the evolving periodic structures in the frequency domain, allowing the model to capture complex changes in periodicity and identify segment-level anomalies. These capabilities directly address the limitations highlighted in Challenge 1 and Challenge 2, thereby explaining the overall advantage of CS-LSTMs over prior methods.

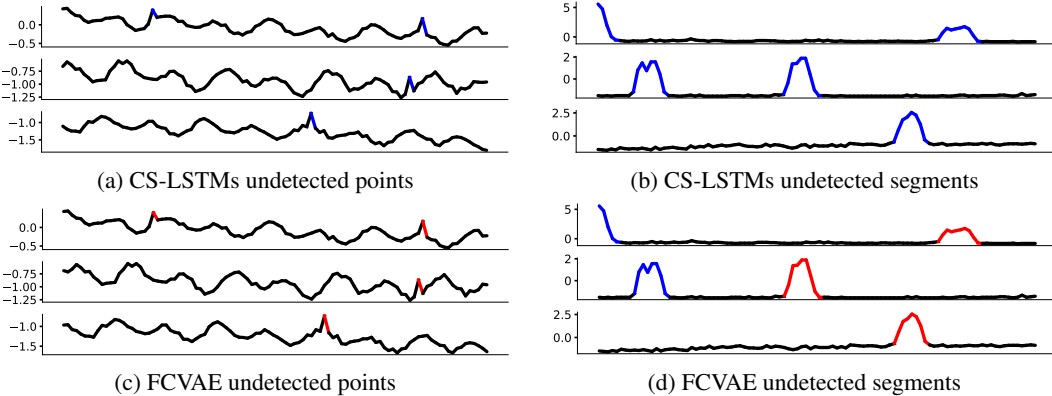

(a) CS-LSTMs undetected points  (b) CS-LSTMs undetected segments

(c) FCVAE undetected points  (d) FCVAE undetected segments

Figure 10: Comparison of CS-LSTMs and FCVAE in detecting specific anomaly types. Normal points are shown in black, detected anomalies in blue, and undetected anomalies in red. Subfigures (a) and (b) present the results of CS-LSTMs on Small Point Anomalies and Slowly Rising Anomalies, respectively, while (c) and (d) illustrate the corresponding results from FCVAE.

# F   LOSS FUNCTION

In this section, we investigate the impact of loss function design on model performance, with a particular focus on the application of *NLL* and anomaly masking. We conduct ablation studies comparing commonly used loss functions, including *MAE* and *MSE*, with the *NLL* employed in this work. As shown in Table 9, the results validate the effectiveness of our proposed improvements, significantly enhancing the model's ability to capture normal patterns and accurately detect anomalies.

Table 9: Branch ablation experiments across Yahoo, KPI, and WSD datasets.

| Method | Yahoo | | KPI | | WSD | | NAB | |
|---|---|---|---|---|---|---|---|---|
| | Best | Delay | Best | Delay | Best | Delay | Best | Delay |
| MSE | 0.722 | 0.676 | 0.828 | 0.495 | 0.852 | 0.604 | 0.985 | 0.889 |
| MAE | 0.712 | 0.695 | 0.909 | 0.757 | 0.850 | 0.602 | 0.983 | 0.884 |
| NLL | **0.885** | **0.878** | **0.936** | **0.879** | **0.910** | **0.857** | **0.996** | **0.918** |

We consider the noise-decomposed Negative Log-Likelihood (NLL) loss for a time series of length $n$:

$$\mathcal{D}(\boldsymbol{\mu}, \boldsymbol{\sigma}, \mathbf{x}, \hat{\mathbf{x}}) = \sum_{t=1}^{n} \left[ \log \sigma_t^2 + \frac{\left( x_t \odot \mathrm{mask}_t + \hat{x}_t \odot \tilde{\mathrm{mask}}_t - \mu_t \right)^2}{\sigma_t^2} \right], \qquad \boldsymbol{\sigma} > 0 \qquad (17)$$

where $\mathbf{x}, \hat{\mathbf{x}} \in \mathbb{R}^n$ denote the original and denoised sequences, $\boldsymbol{\mu}, \boldsymbol{\sigma} \in \mathbb{R}^n$ the predicted means and standard deviations, and $\mathrm{mask} \in \{0, 1\}^n$ indicates normal points.

Defining the residual vector

$$\mathbf{r} := \mathbf{x} \odot \mathrm{mask} + \hat{\mathbf{x}} \odot \tilde{\mathrm{mask}} - \boldsymbol{\mu}$$

The loss can be compactly rewritten as

$$\mathcal{D}(\boldsymbol{\mu}, \boldsymbol{\sigma}, \mathbf{x}, \hat{\mathbf{x}}) = \sum_{t=1}^{n} \log \sigma_t^2 + \sum_{t=1}^{n} \frac{r_t^2}{\sigma_t^2}$$

**Optimal mean.** For fixed $\boldsymbol{\sigma}$, the loss is strictly convex in $\boldsymbol{\mu}$, with gradient

$$\frac{\partial \mathcal{D}}{\partial \boldsymbol{\mu}} = -2 \frac{\mathbf{r}}{\boldsymbol{\sigma}^2}$$

Setting this to zero yields the unique global minimizer

$$\boldsymbol{\mu}^\star = \mathbf{x} \odot \mathrm{mask} + \hat{\mathbf{x}} \odot \tilde{\mathrm{mask}} \qquad (18)$$

Intuitively, the optimal mean corresponds to the hybrid sequence in which original values are retained at normal points and replaced by denoised estimates at abnormal points.

**Optimal standard deviation.** For fixed $\boldsymbol{\mu}$, each $\sigma_t$ can be optimized independently. Differentiating with respect to $s_t = \sigma_t^2$ and solving $\frac{\partial}{\partial s_t}(\log s_t + r_t^2/s_t) = 0$ gives

$$\sigma_t^\star = |r_t|$$

Hence, the optimal standard deviation at each time step equals the absolute residual between the hybrid sequence and the predicted mean. In practice, a small lower bound $\sigma_{\min} > 0$ is imposed to prevent degenerate solutions.

**Extension to composite loss.** For the combined seasonal and contextual loss

$$L = L_s + L_c = \mathcal{D}(\boldsymbol{\mu}_s, \boldsymbol{\sigma}_s, \mathbf{x}, \hat{\mathbf{x}}) + \mathcal{D}(\boldsymbol{\mu}_c, \boldsymbol{\sigma}_c, \mathbf{x}, \hat{\mathbf{x}})$$

the global optima $(\boldsymbol{\mu}_s^\star, \boldsymbol{\sigma}_s^\star)$ and $(\boldsymbol{\mu}_c^\star, \boldsymbol{\sigma}_c^\star)$ are obtained independently using the same elementwise formulae.

**Convergence remark.** The loss is smooth for $\sigma_t \geq \sigma_{\min} > 0$, strictly convex in $\boldsymbol{\mu}$, and coercive in $\boldsymbol{\sigma}$. Standard first-order optimization methods with appropriate step sizes converge to stationary points, which coincide with the elementwise global minimizers. Consequently, the masked Gaussian NLL admits well-defined and attainable optima under the hybrid masking scheme.

## G TRAINING EFFICIENCY EVALUATION

To further evaluate training efficiency, we conducted additional experiments recording resource consumption for training one epoch under the same dataset and experimental settings as the baselines, including overhead introduced by windowing and FFT operations. The results shown in Table 10 demonstrate that CS-LSTMs maintain competitive training efficiency compared to strong baselines. Despite the additional computations for Noise-Decomposition processing, the lightweight design of both C-LSTM and S-LSTM branches, together with efficient batching strategies, ensures that the overall training time per epoch remains advantageous.

Table 10: Training efficiency comparison of CS-LSTMs and baseline models.

| Model | CPU Time | GPU Time | GPU Memory |
|---|---|---|---|
| CS-LSTMs | 210 s | 39 s | 71 MB |
| FCVAE | 22 s | 16 s | 68 MB |
| Informer | 530 s | 530 s | 316 MB |
| Anotransformer | 483 s | 483 s | 333 MB |
| TFAD | 1666 s | 75 s | 4048 MB |

## H  ADDITIONAL EVALUATION ON MEDICAL AND ENVIRONMENTAL DATASETS

To further address the reviewer's concern regarding the generalizability of the proposed method, we conducted additional experiments on two widely used non-IT univariate time series datasets from the UCR Archive: ECG (medical) and CIMIS44AirTemperature (environmental temperature).

Although CS-LSTMs are primarily motivated by AIOps scenarios where univariate anomaly detection is a central challenge, our method is designed to model both periodicity evolution and local contextual variations, which naturally extend to diverse time series domains. Consistent with results on IT monitoring datasets, CS-LSTMs continue to outperform strong baselines on these two new domains, demonstrating strong cross-domain robustness. The detailed performance comparison is shown in Table 11.

Table 11: Performance comparison on ECG and CIMIS44AirTemperature. **F1** denotes the best F1 score, **P** denotes Precision, and **R** denotes Recall.

| Method | ECG | | | | | | CIMIS44AirTemperature | | | | | |
|---|---|---|---|---|---|---|---|---|---|---|---|---|
| | **Best** | | | **Delay** | | | **Best** | | | **Delay** | | |
| | **F1** | **P** | **R** | **F1** | **P** | **R** | **F1** | **P** | **R** | **F1** | **P** | **R** |
| CS-LSTMs | **0.918** | 0.849 | 1.000 | **0.823** | 0.699 | 1.000 | **0.950** | 0.910 | 0.995 | 0.809 | 0.883 | 0.746 |
| FCVAE | 0.914 | 0.842 | 0.819 | 0.735 | 0.820 | 0.667 | 0.850 | 0.741 | 0.995 | **0.850** | 0.741 | 0.995 |
| Anomaly-Transformer | 0.594 | 0.741 | 0.496 | 0.153 | 0.218 | 0.118 | 0.810 | 0.805 | 0.815 | 0.313 | 0.638 | 0.207 |
| TFAD | 0.835 | 0.783 | 0.894 | 0.713 | 0.684 | 0.745 | 0.772 | 0.718 | 0.835 | 0.756 | 0.733 | 0.780 |
| Informer | 0.736 | 0.682 | 0.799 | 0.715 | 0.658 | 0.783 | 0.842 | 0.768 | 0.932 | 0.836 | 0.771 | 0.913 |
| KAN-AD | 0.916 | 0.846 | 1.000 | 0.797 | 0.991 | 0.667 | 0.842 | 0.855 | 0.870 | 0.842 | 0.815 | 0.870 |
| AnoTransfer | 0.695 | 0.803 | 0.613 | 0.545 | 0.525 | 0.567 | 0.562 | 0.752 | 0.449 | 0.540 | 0.608 | 0.486 |
| VQRAE | 0.316 | 0.524 | 0.226 | 0.145 | 0.185 | 0.119 | 0.504 | 0.688 | 0.398 | 0.494 | 0.672 | 0.391 |
| DONUT | 0.513 | 0.420 | 0.391 | 0.445 | 0.405 | 0.494 | 0.255 | 0.410 | 0.185 | 0.255 | 0.410 | 0.185 |
| SRCNN | 0.740 | 0.705 | 0.779 | 0.636 | 0.602 | 0.674 | 0.285 | 0.310 | 0.264 | 0.212 | 0.246 | 0.186 |
| SPOT | 0.412 | 0.932 | 0.264 | 0.136 | 0.885 | 0.074 | 0.432 | 0.560 | 0.352 | 0.448 | 0.542 | 0.382 |

## I  THE USE OF LLMS

The authors affirm that throughout this study, LLMs were solely for the translation and polishing of the manuscript. LLMs were not involved in the literature search for the related work section or in the formulation of the research idea. The authors take full responsibility for this declaration.

