# OpenReview forum: "Contextual and Seasonal LSTMs for Time Series Anomaly Detection"
_ICLR.cc/2026/Conference — ICLR 2026 Poster_

### Official Review · Reviewer_Bshn · 2025-10-28

**Soundness:** 2
**Presentation:** 2
**Contribution:** 2
**Rating:** 4
**Confidence:** 2

**Summary:**

The paper identifies key shortcomings of prior approaches to univariate time-series (UTS) anomaly detection and proposes CS-LSTM. This dual-branch architecture explicitly separates long-term seasonal dynamics from short-term contextual fluctuations. By assigning distinct roles to the two LSTM branches, the method aims to detect both *small point anomalies* and *slowly rising interval anomalies.* Across several benchmarks, the approach is reported to outperform prior methods in both accuracy and runtime.

**Strengths:**

- **Clear problem framing**. This paper is, to my knowledge, the first to explicitly explain the two challenges of *small point anomalies* and *slowly rising anomalies* in UTS, and to analyze why popular reconstruction- and prediction-based baselines struggle with them.
- **Probabilistic predictions**. Instead of deterministic outputs, the model predicts $\mu$ and $\sigma$, allowing $\sigma$ to act as a data-driven tolerance band and enabling a principled anomaly score.
- **Architectural novelty with complementary roles**. The design assigns seasonal/long-term dynamics to the S-LSTM and contextual/short-term dynamics to the C-LSTM, yielding strong empirical performance. The dual-branch decomposition is simple, interpretable, and appears novel in this setting.

**Weaknesses:**

- **Generalization beyond the targeted anomaly types.** The evaluation focuses on cases where the two targeted anomaly types are present. It remains unclear how the model behaves when anomalies fall outside these categories (e.g., regime shifts without seasonality, changes in variance, or adversarial bursts).
- **Scope limited to univariate data.** Many real-world series are multivariate. It is not apparent how this UTS-specific design would extend to multivariate settings.
- **Hyperparameter sensitivity.** The method introduces several windowing hyperparameters (e.g., $w_{s}$, $w_{c}$, $k$). The relative importance of these choices, guidance for tuning, and robustness across datasets and tasks are not fully clarified; optimal values are likely to vary substantially across domains, which may complicate deployment.

**Questions:**

- **Training efficiency.** Table 4 reports inference latency. How does training time compare to strong baselines under the same hardware and data conditions (including windowing/FFT overhead)?
- **Mask estimation.** Please describe the procedure to construct the mask used in the masked Gaussian NLL in full detail (initialization, update rule, thresholds/criteria).

---

> ### Author Response · Authors · 2025-11-22
> **Response to Reviewer Bshn (Q1-Q2)**
>
> **Q1: Training efficiency. Table 4 reports inference latency. How does training time compare to strong baselines under the same hardware and data conditions (including windowing/FFT overhead)?**
>
> A1:  Thanks. We conducted additional experiments to evaluate training efficiency. Specifically, we recorded resource consumption for training one epoch under the same dataset and experimental settings as the baselines, including the overhead introduced by windowing and FFT operations. The results show that CS-LSTMs maintain competitive training efficiency compared to strong baselines. Despite the additional computations for Noise-Decomposition processing, the lightweight design of both C-LSTM and S-LSTM branches, together with efficient batching strategies, ensures that the overall training time per epoch still maintains a substantial advantage.
>
> **Training efficiency comparison is as follows**:
> | Model | CPU Time (s) | GPU Time (s) | GPU Memory |
> |-|-|-|-|
> | CS-LSTMs | 210 | 39 | 71 MB |
> | FCVAE | 22 | 16 | 68 MB |
> | Informer | 530 | 530 | 316 MB |
> | Anotransformer | 483 | 483 | 333 MB |
> | TFAD | 1666 | 75 | 4048 MB |
>
> ---
> **Q2: Mask estimation. Please describe the procedure to construct the mask used in the masked Gaussian NLL in full detail (initialization, update rule, thresholds/criteria).**
>
> A2:  Thanks. We first note that, as mentioned in the introduction, anomalies in real-world tasks are extremely sparse, typically accounting for only 0.5%–5% of the data, and the ground truth normal values at anomaly positions are unknown. This leads to two consequences:
> (1) The majority of values are normal, allowing the model to naturally learn normal patterns in the time series, which is desirable.
>
> (2) When predicting the normal range of future points during training, using the raw sequence as labels may bias predictions—normal points will yield accurate ranges, but anomaly points may skew the predicted range toward abnormal values. This is the issue we aim to mitigate.
> To address this, we introduce a noise decomposition and mask mechanism, supporting both supervised and unsupervised modes:
>
> **Supervised mode**: We use the original anomaly labels of the time series to generate the mask. Suppose the predicted sequence length is $N_p$. During training, we take the logical negation of the corresponding segment's labels to obtain a mask of length $N_p$, where positions corresponding to normal points are 1 and anomaly points are 0. After the model outputs the predicted range, we compute the loss using Equation 7, where the mask replaces the values at anomaly positions in the original sequence $x$ with the denoised sequence $\hat x$. This ensures that minimizing the loss encourages the predicted range to align with normal values. Given the prediction window length $N_p$, we construct the mask from anomaly labels:
>
> $$
> \text{mask}[t] = 1\quad \text{if } x_t \text{ is normal else} \quad 0,
> $$
> $$
> \widetilde{\text{mask}}[t] = 1 - \text{mask}[t].
> $$
>
> Let $\hat{x}$ denote the denoised sequence obtained via wavelet-based noise decomposition. We replace anomalous ground-truth values using:
>
> $$x^{\text{target}} = \text{mask} \odot x + (1 - \text{mask}) \odot \hat{x}.$$
>
> This prevents anomalous points from distorting the learned normal patterns.
>
> **Unsupervised mode**: To enhance robustness, we manually inject anomalous points or segments and label these positions as 1, with other positions labeled as 0. The mask is then generated in the same manner as in the supervised mode and used to optimize the loss function similarly.
> This design ensures that the model learns to predict normal ranges accurately while mitigating the influence of anomalies during training.
>
> $$\text{mask}_{\text{unsup}}[t] = 0 \quad\text{if } t \text{ is an injected anomaly else} \quad 1,$$
> The same masking process is then used to construct $x^{\text{target}}$.
>
> **Loss application**: After the S-LSTM and C-LSTM branches produce the predicted normal range $(\mu, \sigma)$,  we compute the loss in Eq. 7 using the masked target:
> $$
> \mathcal{D}(\mu, \sigma, x, \hat{x}) =
> \log \sigma^2 + \frac{(x \odot \text{mask} + \hat{x} \odot \tilde{\text{mask}} - \mu)^2}{\sigma^2}.$$
> Normal timestamps contribute standard regression constraints, while anomalous values are replaced by the denoised estimates $\hat{x}$.  This design prevents anomalies from biasing the model and is validated by our ablation study (Table 5).

---

> ### Author Response · Authors · 2025-11-22
> **Response to Reviewer Bshn (W1)**
>
> **W1: Generalization beyond the targeted anomaly types. The evaluation focuses on cases where the two targeted anomaly types are present. It remains unclear how the model behaves when anomalies fall outside these categories (e.g., regime shifts without seasonality, changes in variance, or adversarial bursts).**
>
> A1: We thank the reviewer for raising this concern. We would like to clarify that our work is not specifically designed or optimized for the two challenging anomaly types (small point anomalies and slowly rising anomalies). Our discussion of these two types serves only to illustrate the weaknesses of existing methods, rather than to define the scope of our model. We further elaborate as follows:
>
> **1. Dataset coverage**: The datasets used in our experiments naturally contain a wide variety of anomaly types beyond the two targeted ones.
> The four real-world benchmark datasets (Yahoo, KPI, WSD, NAB) include anomalies such as:  mean/regime shifts, variance changes, bursts/spikes, long contextual anomalies, non-seasonal trends, and multiple types of point anomalies. These anomalies are inherent in the datasets, not artificially selected or constructed. Our model achieves state-of-the-art performance across all datasets (Table 1), which empirically demonstrates its generalization to diverse anomaly types.
>
> **2. Model design**: CS-LSTMs are inherently capable of handling multiple anomaly patterns and do not rely on specific anomaly types.
> As a predictive model, CS-LSTMs estimate the normal range of future points and compare them with the observed values to detect anomalies.
> This makes it sensitive to points outside the overall sequence distribution. Key design factors supporting generalization:
>   - Sequence values as covariates → capture mean/regime shifts and bursts/spikes
>   - Combination of long-term & short-term frequency information → detect variance changes and segment anomalies
>   - LSTM backbone → models trends with or without seasonality
>   - S-LSTM branch → easily detects periodic anomalies
>
> **3. Empirical validation of generalization**:
> Although our paper highlights the advantages of the two challenging anomaly types, the model’s overall superior performance across all benchmark datasets demonstrates that it does not rely on specific anomaly patterns and is robust to diverse anomaly types.
> Furthermore, in cross-dataset transfer experiments (Table 3), the model maintains strong performance under unknown data distributions and previously unseen anomaly structures, further confirming its generalization ability to **non-targeted anomaly types**.

---

> ### Author Response · Authors · 2025-11-22
> **Response to Reviewer Bshn (W2-W3)**
>
> **W2: Scope limited to univariate data. Many real-world series are multivariate. It is not apparent how this UTS-specific design would extend to multivariate settings.**
> A2:  Thank you for the question. CS-LSTMs are originally designed for **univariate time series**, as most anomaly detection tasks in AIOps monitoring systems focus on single KPIs. Many recent studies also target univariate settings, including FCVAE (Wang et al., 2024), TFAD (Zhang et al., 2022).
>
> Extending CS-LSTMs to **multivariate time series** is feasible but would require modifications to the model architecture. In the current design, the two branches separately learn: the **evolution of periodicity** and **contextual variations**. For a single sequence, without explicitly modeling dependencies across multiple channels. To handle multivariate data, one could incorporate: a **convolutional encoder** to capture global inter-channel correlations, followed by LSTMs to model temporal dependencies across channels. However, this extension is beyond the scope of the current work. We consider it a promising direction for future research.
>
> ---
> **W3: Hyperparameter sensitivity. The method introduces several windowing hyperparameters (e.g., total window length, seasonal window size, contextual window size). The relative importance of these choices, guidance for tuning, and robustness across datasets and tasks are not fully clarified; optimal values are likely to vary substantially across domains, which may complicate deployment.**
> A3: Thank you for the question. CS-LSTMs adopt different hyperparameters across datasets and tasks mainly due to differences in the intrinsic information density of each dataset. For example, Yahoo uses a 1-hour sampling interval, NAB uses 5 minutes, while WSD and AIOps use 1 minute. Many recent studies—such as FCVAE (Wang et al., 2024) and TFAD (Zhang et al., 2022)—also employ dataset-specific hyperparameters for similar reasons. In practice, only **three hyperparameters** need to be set:
> - **total_window_size**: the input sequence length for inference
> - **seasonal_window_size**: the segmentation window size for S-LSTM
> - **contextual_window_size**: the segmentation window size for C-LSTM
>
> Typically, total_window_size and contextual_window_size can be determined from seasonal_window_size.
> In real-world settings, simple prior knowledge is sufficient to estimate the approximate period of a time series (e.g., half-day, daily, or hourly cycles), combined with overlapping windows. This is exactly how we configure all public dataset experiments and does not introduce a significant tuning burden. For stable learning, we use the following practical rules:
> - **To capture periodicity evolution**: Total_window_size = m $×$ seasonal_window_size, with $m ∈ (5, 7)$.
> - **To capture contextual variations**: Contextual_window_size = seasonal_window_size // n, with $n ∈ (5, 7)$.
>
> These settings consistently yield good performance across datasets. Moreover, as shown in Figure 7, CS-LSTMs are highly robust to window-size choices: within a reasonable range, window variations change performance by only about 2%. Therefore, CS-LSTMs do not impose significant complexity in deployment and remain robust even with coarse-to-fine hyperparameter tuning.

---

> ### Author Response · Authors · 2025-11-26
> **Response to Reviewer Bshn**
>
> Dear Reviewer,
>
> We'd appreciate it if you'd let us know if our response has addressed your concerns.
>
> Thanks!

---

> > ### Comment · Reviewer_Bshn · 2025-11-27
> > **Reply**
> >
> > I believe I now have a solid understanding of the proposed algorithm. However, I remain unconvinced regarding the degree of architectural novelty: the design still appears largely engineering-oriented, and it is not clear to me that the resulting contribution is sufficiently substantial to warrant acceptance at this venue. Moreover, I consider scalability to multivariate settings to be an important issue, which is not adequately addressed in the current version. For these reasons, I choose to keep my overall score unchanged, with my confidence remaining at 2.

---

### Official Review · Reviewer_S6ae · 2025-10-30

**Soundness:** 3
**Presentation:** 3
**Contribution:** 3
**Rating:** 6
**Confidence:** 4

**Summary:**

The paper introduces CS-LSTMs, a dual-branch prediction framework for detecting subtle anomalies in univariate time series (UTS). It consists of two complementary LSTM branches, S-LSTM (Seasonal branch) captures evolving periodicity via wavelet-based time–frequency representations, and C-LSTM (Contextual branch) models short-term dependencies and local dynamics in the time domain.

A wavelet transform first decomposes the signal into trend, seasonal, and noise components, allowing each branch to process denoised inputs. Their outputs are fused for final anomaly prediction, optimized by a masked probabilistic loss that focuses learning on normal patterns while suppressing anomalous effects.

Across multiple benchmarks, CS-LSTMs consistently outperform reconstruction- and prediction-based baselines. Ablation and transfer experiments verify that the dual-branch design and noise-decomposition mechanism jointly enhance robustness and generalization. The approach effectively overcomes prior models’ difficulty in detecting small point and gradually rising anomalies by integrating time- and frequency-domain representations and modeling the evolution of periodicity.

**Strengths:**

- The separation of seasonal and contextual learning provides complementary modeling of periodic and local behaviors, enabling precise detection of small or slowly evolving anomalies.
- The use of wavelet-based decomposition allows simultaneous handling of non-stationarity and noise suppression, improving interpretability and robustness compared with pure time-domain models.
- Demonstrates consistent superiority over SOTA baselines across several datasets, supported by transferability and ablation experiments showing each module’s distinct contribution.
- Addresses a realistic gap in anomaly detection—capturing weak, evolving irregularities that traditional reconstruction or forecasting models often miss.

**Weaknesses:**

- Key mechanisms—especially the noise-decomposition pipeline, fusion process, and masked loss formulation—lack detailed mathematical description and ablation justification.
- The paper does not analyze how the method scales to multivariate time series or handles cross-variable dependencies.
- Efficiency metrics are presented in relative terms without absolute computational cost (e.g., GPU time, memory).

**Questions:**

- How exactly does the wavelet-based noise decomposition feed into the dual-branch architecture? Need clear data flow and mathematical formulation.
- How are mask and ~mask generated and applied in the loss function to handle anomalies during training?
- How is ‘evolution of periodicity’ defined and quantified within the S-LSTM branch?
- Why does the paper choose LSTM over Transformer-based alternatives (Informer, Autoformer) for subtle anomaly modeling?
- Can CS-LSTMs handle multivariate or cross-correlated series, and what modifications would be required?
- Recent works such as Dual-TF (Nam et al., 2024), TFAD (Zhang et al., 2022), FAD (Li et al., 2023), and WaveletAE (Zhao et al., 2020) also integrate time–frequency representations for anomaly detection. How do CS-LSTMs differ from these dual-TF or frequency-aware models in terms of architectural design, ability to capture evolving periodicity, and computational efficiency?

---

> ### Author Response · Authors · 2025-11-22
> **Response to Reviewer S6ae (Q1-Q2)**
>
> **Q1: How exactly does the wavelet-based noise decomposition feed into the dual-branch architecture? Need clear data flow and mathematical formulation.**
>
> A1:  We thank the reviewer for pointing out the need for a clearer description of how the wavelet-based noise decomposition interacts with the dual-branch architecture. Below, we provide the complete data flow and the corresponding mathematical formulation.
>
> First, we clarify that the wavelet-based noise decomposition is used only during training, and its output $\hat{x}$ is used only in the loss computation. The decomposition module is not used during inference, which contributes to the high runtime efficiency of CS-LSTMs.
>
> **Data Flow and Mathematical Formulation**
>
> Let the raw input sequence be $x \in \mathbb{R}^{N}$.  During training, we first obtain a denoised estimate through wavelet decomposition: $\hat{x} = \mathrm{Noise\text{-}Decomposition}(x).$
>
> **Seasonal Branch (S-LSTM)**
>
> We segment $x$ into non-overlapping windows of size $W_s$ to get $\mathrm{input}_S \in \mathbb{R}^{\frac{N}{W_s} \times W_s},$ which are fed into the S-LSTM to model seasonal evolution patterns: $(\mu_S, \sigma_S) = \mathrm{S\text{-}LSTM}(\mathrm{input}_S).$
>
> **Context Branch (C-LSTM)**
>
> We segment $x$ into overlapping windows of size $W_c$ with stride $\mathrm{step} < W_c$.  In practice, we set: $\mathrm{step} = \frac{W_c}{2},$ chosen heuristically to balance fine-grained context and information coverage. The overlapping segments are:
> $\mathrm{input}_C \in \mathbb{R}^{\left(\frac{W_s - W_c}{\mathrm{step}} + 1\right) \times W_c}$, and the C-LSTM predicts:
> $(\mu_C, \sigma_C) = \mathrm{C\text{-}LSTM}(\mathrm{input}_C).$
>
> **Loss Computation Using the Denoised Series $\hat{x}$:**
>
> The denoised series is incorporated only during loss computation:
>
> $$
> \mathcal{D}(\mu, \sigma, x, \hat{x})
> = \log \sigma^{2} + \frac{\left( x \odot \text{mask} + \hat{x} \odot \tilde{\text{mask}} - \mu \right)^2}{\sigma^{2}},
> $$
>
> $$\mathcal{L} = \mathcal{D}_S + \mathcal{D}_C,$$
>
> where the two terms follow the formulation in the original paper.
> This stabilizes training by reducing the influence of anomalies and high-frequency noise.
>
> Inference Stage
> During inference:
> - No wavelet decomposition is performed.
> - The model processes only the raw sequence $x$ through the S-LSTM and C-LSTM branches.
> - The combined output yields the predicted normal range for the next point.
> - An anomaly is detected by comparing the real value with the predicted normal range.
> Thus, the noise decomposition module is fully bypassed during inference, explaining the high computational efficiency of CS-LSTMs.
>
> ---
> **Q2: How are mask and ~mask generated and applied in the loss function to handle anomalies during training?**
>
> A2: Thanks. Mask and ~mask generation and their role in the loss. Real-world AIOps time series contain extremely sparse anomalies (typically 0.5%–5%), and the true normal values at anomalous timestamps are unknown. Using the raw sequence directly as regression targets would bias the predicted normal range toward anomalous values. To address this, we introduce a **wavelet-based noise decomposition** and a **masking mechanism**.
>
> **Supervised setting**: Given the prediction window length $N_p$, we construct the mask from anomaly labels:
>
> $$
> \text{mask}[t] = 1\quad \text{if } x_t \text{ is normal else} \quad 0,
> $$
> $$
> \widetilde{\text{mask}}[t] = 1 - \text{mask}[t].
> $$
>
> Let $\hat{x}$ denote the denoised sequence obtained via wavelet-based noise decomposition. We replace anomalous ground-truth values using:
>
> $$x^{\text{target}} = \text{mask} \odot x + (1 - \text{mask}) \odot \hat{x}.$$
>
> This prevents anomalous points from distorting the learned normal patterns.
>
> **Unsupervised setting**: When no labels are available, we inject synthetic point or segment anomalies during training.  Injected anomalies receive mask value 0:
> $$\text{mask}_{\text{unsup}}[t] = 0 \quad\text{if } t \text{ is an injected anomaly else} \quad 1,$$
> The same masking process is then used to construct $x^{\text{target}}$.
>
> **Loss application**
>
> After the S-LSTM and C-LSTM branches produce the predicted normal range $(\mu, \sigma)$,  we compute the loss in Eq. 7 using the masked target:
> $$
> \mathcal{D}(\mu, \sigma, x, \hat{x}) =
> \log \sigma^2 + \frac{(x \odot \text{mask} + \hat{x} \odot \tilde{\text{mask}} - \mu)^2}{\sigma^2},$$
> Normal timestamps contribute standard regression constraints, while anomalous values are replaced by the denoised estimates $\hat{x}$.  This design prevents anomalies from biasing the model and is validated by our ablation study (Table 5).

---

> ### Author Response · Authors · 2025-11-22
> **Response to Reviewer S6ae (Q3-Q5)**
>
> **Q3: How is ‘evolution of periodicity’ defined and quantified within the S-LSTM branch?**
>
> A3:  Thanks. We interpret the **“evolution of periodicity”** as the temporal variation of periodic patterns in a time series, as illustrated in Figure 4. Formally, consider a hypothetical periodic function $f(X)$ satisfying $f(X) = f(X + \Delta t).$ Suppose a time series $(X,y)$ follows $f(X)$in the interval
> $[t, t + \Delta t)$, $f(X)+1$ in $[t, t + 2\Delta t)$, and $f(X)+2$ in $[t, t + 3\Delta t)$. Clearly, this series does not exhibit strict periodicity, but it can be represented as a composition of a base periodic function $f(X)$ and a slowly-varying function $g(X)$. We define $g(X)$ as the evolution of periodicity, which is the pattern that the S-LSTM branch aims to learn.
>
> Specifically:
>
> **1. Frequency-domain representation:**
> By transforming a time series into the frequency domain, its periodic structure appears as **stable and concentrated spectral peaks**, which are much clearer and more robust than observing raw waveforms in the time domain. Even when adjacent segments exhibit phase shifts, their **magnitude spectra** remain highly consistent, high- and low-frequency components correspond one-to-one without being affected by phase misalignment. This allows periodic structures to be directly compared across segments, avoiding the complexity of temporal alignment. Meanwhile, noise and non-periodic components typically manifest as dispersed high-frequency disturbances in the spectrum, making them easier to separate from true periodic components. As a result, the frequency-domain representation makes periodic patterns more salient and easier to model. These properties enable the S-LSTM branch to more effectively capture periodic patterns and their temporal evolution.
>
> **2. Learning with S-LSTM:**
> These consecutive frequency tensors are fed into the LSTM, which captures the temporal dynamics of both high- and low-frequency components, effectively modeling $g(X)$, the evolution of periodicity.
>
> In summary, the S-LSTM branch **learns the temporal evolution of underlying periodic patterns** by analyzing frequency-domain features over consecutive time segments.
>
> ---
> **Q4: Why does the paper choose LSTM over Transformer-based alternatives (Informer, Autoformer) for subtle anomaly modeling?**
>
> A4:  Thank you for the question. Transformer-based methods are primarily designed for multivariate time series (MTS) tasks. For instance, Anomaly-Transformer and DCDector target anomaly detection in MTS, while Informer and Autoformer are designed for MTS forecasting. These methods perform well when the data is high-dimensional and rich in temporal patterns.
>
> However, as discussed in our paper, in univariate time series (UTS) tasks, each time step contains only a single value. This setting presents several challenges:
>
> - **Sparse data points**: Limited temporal context at each step.
> - **Homogeneous distribution**: Lack of diverse signals to exploit.
> - **Simple patterns**: Most variations are local rather than global.
>
> Due to these characteristics, Transformer-based models struggle to learn stable patterns. Our experimental results (Table~1) confirm this observation.
>
> In contrast, LSTM naturally captures:
>
> - **Local dependencies**: Leveraging sequential context effectively.
> - **Smooth transitions**: Modeling gradual changes between time steps.
> - **Local pattern memory**: Retaining essential information for anomaly detection.
>
> Since univariate time series are dominated by local patterns, LSTMs are often more suitable. This explains why, despite the widespread application of Transformers across many domains, simpler sequential models like LSTM remain widely used in time series tasks.
>
> ---
> **Q5: Can CS-LSTMs handle multivariate or cross-correlated series, and what modifications would be required?**
>
> A5:  Thanks. CS-LSTM is primarily designed for univariate time series, as in AIOps monitoring systems, most anomaly detection tasks are performed on individual KPIs. This focus is consistent with recent studies, such as FCVAE (Wang et al., 2024), TFAD (Zhang et al., 2022), which also target univariate anomaly detection.
> Extending CS-LSTM to multivariate time series is feasible but would require modifications to the model architecture. In the current design, the two branches separately learn the evolution of periodicity and the contextual changes in the time series, without modeling inter-channel relationships. To handle multivariate or cross-correlated series, one could:
> - Introduce a **convolutional encoder** to capture global correlations across all channels.
> - Feed the encoded representations into the LSTM to learn temporal dependencies.
> However, this extension is beyond the scope of our current task definition. We consider it a promising direction for future work.

---

> ### Author Response · Authors · 2025-11-22
> **Response to Reviewer S6ae (Q6)**
>
> **Q6: Recent works such as Dual-TF (Nam et al., 2024), TFAD (Zhang et al., 2022), FAD (Li et al., 2023), and WaveletAE (Zhao et al., 2020) also integrate time–frequency representations for anomaly detection. How do CS-LSTMs differ from these dual-TF or frequency-aware models in terms of architectural design, ability to capture evolving periodicity, and computational efficiency?**
>
> A6:  We thank the reviewer for pointing out these relevant works. The key differences between CS-LSTMs and prior time–frequency models are summarized as follows:
> - **WaveletAE**: WaveletAE first applies **multi-level discrete wavelet decomposition (MDWD)** to extract multi-scale time–frequency features. The wavelet detail coefficients are concatenated with the original signal and fed into a CNN–LSTM architecture, where the convolutional encoder captures global inter-channel correlations and the LSTM encoder models temporal dynamics. This design effectively learns complex inter-channel relationships and multi-scale dynamic patterns.
> - **TFAD**: TFAD jointly models **time-domain features and frequency-domain features** obtained via short-time Fourier transform (STFT). Time-domain features help capture abrupt, pointwise anomalies, while frequency-domain features reveal changes in spectral components caused by anomalies. Anomaly scores are computed based on window-level frequency consistency and pointwise reconstruction error in the time domain.
> - **Dual-TF**: Dual-TF addresses the mismatch in granularity between **pointwise time-domain** and **windowed frequency-domain** signals. It introduces a **nested sliding window** mechanism with dual reconstructions (time & frequency). An outer window is used for time-domain reconstruction, while multiple inner windows within each outer window allow generating frequency-domain views for each data point. This improves upon TFAD by providing multiple frequency perspectives per time point.
> - **FAD**: We were unable to find this reference.
> - **CS-LSTMs**: In contrast, our method is the first, to the best of our knowledge, to perform anomaly detection via **two branches that simultaneously model evolving periodicity and local variations**. Both branches receive inputs that contain **time-domain and frequency-domain information**, albeit at different scales:
>   - C-LSTM:
>   Utilizes partially overlapping windows to capture trends in frequency-band energy changes within small windows, while incorporating covariates from the time domain to predict future values.
>   - S-LSTM:
>   Performs FFT-based frequency alignment across segments to capture periodic patterns, and then uses LSTM to model the **evolution of periodicity** over time, together with time-domain information for prediction.
>
> Furthermore, CS-LSTMs not only leverage time–frequency analysis but also **explicitly explore the temporal evolution of frequency-domain information**. Unlike prior methods that separately use time- and frequency-domain features to evaluate anomalies, CS-LSTMs integrate both in a dual-branch architecture and explicitly model the temporal evolution of frequency-domain features, enabling more precise detection of subtle anomalies and capturing dynamic changes that previous models do not explicitly model.
>
> | Method | Time–Frequency Representation | Core Architecture | Handles Evolving Periodicity? | Key Limitation | Computational Efficiency |
> |--------|-------------------------------|--------------------|-------------------------------|----------------|---------------------------|
> | WaveletAE (Zhao et al., 2020) | Multi-level discrete wavelet decomposition (MDWD) | CNN + LSTM | No. Wavelet coefficients are static per level | Focuses on multi-scale decomposition, not temporal evolution of periodicity | Requires multi-level wavelet decomposition and CNN-LSTM; medium-high computational cost |
> | TFAD (Zhang et al., 2022) | STFT-based time–frequency fusion | Temporal encoder + Frequency encoder | No. Frequency-domain features used only for reconstruction consistency | Time and frequency signals are processed separately without modeling evolution | STFT plus dual encoders; additional frequency computations slow down training moderately |
> | Dual-TF (Nam et al., 2024) | Nested sliding-window STFT | Dual reconstruction (time + frequency) | Partially. Multiple frequency views per point, but no explicit periodic evolution modeling | High computational cost due to nested windows; focuses on granularity alignment | Nested windows greatly increase computation; training and inference are slow |
> | CS-LSTMs (Ours) | FFT-aligned frequency sequences + Local frequency-band energy trends | Dual-branch LSTM (C-LSTM + S-LSTM) | Yes — explicitly models temporal evolution of periodicity and local variations | — | Frequency processing only during training; inference bypasses FFT, making it computationally efficient |

---

> ### Author Response · Authors · 2025-11-22
> **Response to Reviewer S6ae (W1)**
>
> **W1: Key mechanisms—especially the noise-decomposition pipeline, fusion process, and masked loss formulation—lack detailed mathematical description and ablation justification.**
>
> A1: Thank you for the comment. The noise-decomposition and masked-loss formulation in our method functions as a **unified** pipeline, rather than two independent components. Specifically, during training, we apply a mask to replace the anomalous positions in the original sequence $x$ with their denoised counterparts $\hat x$, which are obtained through our noise-decomposition module. The detailed mathematical description is as follows:
>
> $$\{c_A, c_D^{(L)}, \ldots, c_D^{(1)}\} = \text{wavedec}(x, \psi, L),$$
> $$\sigma_i = \frac{\text{median}(|c_D^{(i)}|)}{\Phi^{-1}(0.75)}, \quad
> \lambda_i = \sigma_i \sqrt{2 \log n}, \quad
> \forall i = 1, \ldots, L,$$
> $$\hat{c}_D^{(i)} = \text{sign}(c_D^{(i)}) \cdot \max(|c_D^{(i)}| - \lambda_i,\, 0),
> \quad \forall i = 1, \ldots, L,$$
> $$\hat{x} = \text{waverec}(c_A, \hat{c}_D^{(L)}, \ldots, \hat{c}_D^{(1)}).$$
>
> The loss is then computed between the model’s prediction and this denoised target. This mechanism effectively suppresses the negative influence of anomalies on the optimization process and enables the model to learn a cleaner representation of normal dynamics.
> $$
> \mathcal{D}(\mu, \sigma, x, \hat{x}) =
> \log \sigma^2 + \frac{(x \odot \text{mask} + \hat{x} \odot \tilde{\text{mask}} - \mu)^2}{\sigma^2}.
> $$
> The relevant ablation results are provided in Table 5, where “w/o Noise Decomposition” corresponds to removing this entire pipeline—including both the noise-decomposition and masked-loss process. The clear performance drop demonstrates the necessity of this design.
>
> Regarding the noise-decomposition pipeline itself, we conducted additional experiments comparing our proposed approach with two commonly used alternatives: **pooling decomposition** and **STL decomposition**. The results are summarized in the table below. Our approach consistently outperforms both baselines in anomaly detection accuracy while also providing significantly higher training efficiency. Upon analysis, pooling decomposition is too coarse-grained—fast but ineffective—whereas STL offers finer granularity but suffers from substantial computational overhead. Overall, our noise-decomposition module strikes a better balance between **accuracy** and **efficiency**, yielding the best performance among all compared methods.
>
> The **best F1 score** is shown in the table below:
> | Dataset | Noise Decomposition | Pooling Decomposition | STL Decomposition |
> |-|-|-|-|
> | Yahoo | 0.885 | 0.872 | 0.877 |
> | NAB | 0.996 | 0.988 | 0.989 |
> | WSD | 0.910 | 0.865 | 0.874 |
> | AIOPS | 0.936 | 0.923 | 0.914 |
>
>
> The **delay F1 score** is shown in the table below:
> | Dataset | Noise Decomposition | Pooling Decomposition | STL Decomposition |
> |-|-|-|-|
> | Yahoo   | 0.878|0.860| 0.871|
> | NAB     | 0.918| 0.904 | 0.890 |
> | WSD     | 0.857| 0.812 | 0.816 |
> | AIOPS   | 0.879| 0.871| 0.875 |
>
> Train time per epoch:
> | Method| Time (s) |
> |-|-|
> | Noise Decomposition  | 52.383   |
> | Pooling Decomposition| 59.332   |
> | STL Decomposition    | 458.389  |

---

> ### Author Response · Authors · 2025-11-22
> **Response to Reviewer S6ae (W2-W3)**
>
> **W2: The paper does not analyze how the method scales to multivariate time series or handles cross-variable dependencies.**
>
> A2:  Thank you for the comment. Our CS-LSTM framework is intentionally designed for **univariate time series**, which aligns with the typical setting in AIOps monitoring systems. In practice, most anomaly detection tasks are conducted on **individual KPIs**, and many recent studies follow this paradigm, such as FCVAE (Wang et al., 2024), TFAD (Zhang et al., 2022), and KAN-AD (Zhou et al., 2024).
>
> While extending the model to multivariate inputs is feasible, doing so would require **non-trivial architectural modifications**. Our current framework uses two branches to separately capture periodic evolution and local contextual patterns of a single sequence, without modeling cross-variable dependencies.
>
> To support multivariate time series, one could integrate an additional **convolutional encoder** to extract global inter-variable correlations, followed by LSTM layers to capture temporal dependencies. However, this extension is beyond the scope of our current problem formulation. We agree that adapting CS-LSTM to multivariate settings is a promising direction, and we plan to explore this in future work.
>
> ---
> **W3: Efficiency metrics are presented in relative terms without absolute computational cost (e.g., GPU time, memory).**
> A3: To address the reviewer’s concern, we have added a supplementary experiment reporting the absolute computational cost, including GPU inference time and memory consumption. Specifically, we measure the resource usage under the same dataset and the same number of inference steps (512 iterations) to ensure a fair comparison. The results are summarized in the table below:
>
> **Inference time of 512 iterations:**
> | Method | CPU Time (ms) | GPU Time (ms) | GPU Memory (MB) |
> |--------|---------------|---------------|------------------|
> | CS-LSTMs | 3.82 | 4.62 | 44 |
> | FCVAE | 4.02 | 4.02 | 37 |
> | Informer | 2280 | 2280 | 147 |
> | Anomaly-Transformer | 1539 | 1539 | 130 |
> | TFAD | 40 | 40 | 130 |
>
> **Training time of one epoch**:
> | Model | CPU Time (s) | GPU Time (s) | GPU Memory (MB) |
> |-------|--------------|--------------|----------------|
> | CS-LSTMs | 210 | 39 | 71 |
> | FCVAE | 22 | 16 | 68 |
> | Informer | 530 | 530 | 316 |
> | Anotransformer | 483 | 483 | 333 |
> | TFAD | 1666 | 75 | 4048 |

---

> ### Author Response · Authors · 2025-11-26
> **Response to Reviewer S6ae**
>
> Dear Reviewer,
>
> We'd appreciate it if you'd let us know if our response has addressed your concerns.
>
> Thanks!

---

> > ### Comment · Reviewer_S6ae · 2025-11-28
> >
> > Thank you for the detailed rebuttal. It addresses most of my concerns.
> >
> > However, one further improvement I would suggest is adding quantitative comparisons with recent time-frequency models such as Dual-TF, TFAD, FAD, WaveletAE (mentioned in the review). While the qualitative comparison is appreciated, numerical results would provide stronger evidence of the claimed advantages.

---

> ### Author Response · Authors · 2025-11-29
> **Response to Reviewer S6ae (Q6)**
>
> **Q6: Recent works such as Dual-TF (Nam et al., 2024), TFAD (Zhang et al., 2022), FAD (Li et al., 2023), and WaveletAE (Zhao et al., 2020) also integrate time–frequency representations for anomaly detection. How do CS-LSTMs differ from these dual-TF or frequency-aware models in terms of architectural design, ability to capture evolving periodicity, and computational efficiency?**
>
> **A6: Based on the methods discussed above, we conducted corresponding experiments and comparisons. The results show that our CS-LSTMs achieve the best performance in both accuracy and efficiency.**
>
> **Result on Yahoo dataset:**
> | Method      | Best F1 | Best Precision | Best Recall | Delay F1 | Delay Precision | Delay Recall |
> |-------------|---------|----------------|-------------|----------|------------------|--------------|
> | CS-LSTMs    | 0.885   | 0.919          | 0.853       | 0.878    | 0.915            | 0.845        |
> | Dual-TF     | 0.813   | 0.882          | 0.754       | 0.820    | 0.885            | 0.764        |
> | TFAD        | 0.792   | 0.875          | 0.723       | 0.791    | 0.879            | 0.719        |
> | Wavelet-AE  | 0.252   | 0.476          | 0.171       | 0.113    | 0.152            | 0.090        |
>
> **Result on AIOPS dataset:**
> | Method      | Best F1 | Best Precision | Best Recall | Delay F1 | Delay Precision | Delay Recall |
> |-------------|---------|----------------|-------------|----------|------------------|--------------|
> | CS-LSTMs    | 0.936   | 0.918          | 0.955       | 0.879    | 0.905            | 0.855        |
> | Dual-TF     | 0.888   | 0.902          | 0.874       | 0.725    | 0.693            | 0.760        |
> | TFAD        | 0.752   | 0.684          | 0.834       | 0.680    | 0.650            | 0.714        |
> | Wavelet-AE  | 0.516   | 0.488          | 0.547       | 0.478    | 0.520            | 0.442        |
>
> **Inference time of 512 iterations:**
> | Method     | CPU Time (ms) | GPU Time (ms) | GPU Memory (MB) |
> |------------|---------------|----------------|------------------|
> | CS-LSTMs   | 3.82          | 4.62           | 44               |
> | Dual-TF    | 74            | 74             | >24 GB           |
> | TFAD       | 40            | 40             | 130              |
> | Wavelet-AE | 62            | 62             | 105              |

---

### Official Review · Reviewer_3pdC · 2025-10-31

**Soundness:** 3
**Presentation:** 2
**Contribution:** 2
**Rating:** 4
**Confidence:** 3

**Summary:**

This paper addresses anomaly detection in univariate time series (UTS) by identifying three key challenges that limit existing methods' ability to detect subtle anomalies. The authors propose CS-LSTMs, a dual-branch framework where S-LSTM captures long-term periodic patterns through frequency-domain analysis and C-LSTM models short-term local dynamics. A noise decomposition strategy is introduced to improve robustness against unlabeled anomalies during training. Experimental results on four benchmark datasets demonstrate that CS-LSTMs achieves superior F1 scores compared to ten baseline methods while reducing inference time.

**Strengths:**

- **Systematic analysis of detection failures in existing methods.** The paper identifies why current approaches fail on small point anomalies and slowly rising segment anomalies, attributing this to inadequate integration of local trends with periodic variations.
- **Principled noise decomposition strategy.** The wavelet-based approach effectively filters noise while preserving trend and seasonal components, enabling robust learning of normal patterns despite unlabeled anomalies in training data.
- **Well-motivated dual-branch architecture.** S-LSTM captures long-term periodic evolution via frequency-domain analysis while C-LSTM models short-term local dynamics, providing complementary temporal perspectives that enhance detection capability.
- **Strong empirical results with efficiency gains.** CS-LSTMs consistently outperform ten baselines across four datasets while reducing inference time and using fewer parameters, demonstrating practical deployment value.

**Weaknesses:**

- Equation 6 appears to rely on the function or definition presented later in Equation 7. For clarity and logical coherence, the authors should introduce Equation 7 before Equation 6.
- The paper lacks critical implementation details including specific hyperparameters such as  batch size, number of training epochs,window sizes, decomposition Level L.
- The paper provides limited explanation for why LSTMs are better over other recent architectures (Transformers, state-space models) that have also demonstrated superior performance on time series tasks.
- While Section 5.4.1 shows noise decomposition helps, the paper doesn't compare the MAD against other alternatives (pooling-based decomposition ,Robust STL or recent decomposition approaches).
- All datasets come from web/IT monitoring domains, limiting generalizability claims to other time series applications (medical, financial, environmental).

**Questions:**

- How is the decomposition level L in Equation 2 determined Is it already set as hyperparameters? or is it just automatically derived based on discrete wavelet transform (DWT)?
- In equation 7, there’s Kronecker product in the equation while Appendix E shows element-wise multiplication for the same operation. Which is correct? Couldd the author please clarify and ensure consistent notation.
- In Section 5.4.2, the presentation is confusing. If the top graphs vary seasonal window with fixed context window (and bottom panels do the reverse), why do the fixed window values differ across datasets (e.g., context=4 for Yahoo vs. context=48 for NAB)? Could the author please clarify the experimental setup and using different baseline window sizes across datasets.
- Table 4 reports overall inference time but doesn't break down the wavelet decomposition overhead. How much time does the noise decomposition step consume relative to the dual-branch prediction?

---

> ### Author Response · Authors · 2025-11-22
> **Response to Reviewer 3pdC (Q1-Q2)**
>
> **Q1: How is the decomposition level L in Equation 2 determined? Is it set as a hyperparameter, or automatically derived by the DWT?**
>
> A1: Thank you for the question.
> The decomposition level L in Equation 2 is neither arbitrarily chosen nor automatically set by the DWT library. Instead, it is analytically determined based on the length of the input sequence.
> Specifically, for an input window of length N, prior studies in wavelet-based time series analysis suggest that the decomposition depth should approximately satisfy:
> $L\approx\log_2(N)$.
> This principle is supported by classical wavelet theory and wavelet-based time series analysis literature [1–4].
> Following this principle, our method sets:
> $L = \lfloor \log_2(N) \rfloor$.
> This provides a balance between detail preservation and computational efficiency.
> Therefore, L is a derived hyperparameter computed from the window size, rather than a manually tuned value or one automatically inferred by the DWT implementation.
>
> **Reference**:
> 1. Akansu A N, Haddad R A. Multiresolution signal decomposition: transforms, subbands, and wavelets[M]. Academic press, 2001.
> 2. Percival D B, Walden A T. Wavelet methods for time series analysis[M]. Cambridge university press, 2000.
> 3. Yuan B, Wang C, Luo C, et al. WaveletAE: A wavelet-enhanced autoencoder for wind turbine blade icing detection[J]. arXiv preprint arXiv:1902.05625, 2019.
> 4. Misiti M, Misiti Y, Oppenheim G, et al. Wavelet toolbox[J]. The MathWorks Inc., Natick, MA, 1996, 15: 21.
>
> ---
> **Q2: In Equation 7, there is a Kronecker product in the equation, while Appendix E shows element-wise multiplication for the same operation. Which is correct? Could the authors clarify and ensure consistent notation?**
>
> A2: Thank you for pointing this out. We apologize for the confusion caused by the inconsistent notation.
> In both Equation 7 and Appendix E, the intended operation is element-wise multiplication, which is also the operation used in our actual implementation.
> In the revised manuscript, we have unified the notation across all sections to ensure consistency and avoid ambiguity.

---

> ### Author Response · Authors · 2025-11-22
> **Response to Reviewer 3pdC (Q3-Q4)**
>
> **Q3: In Section 5.4.2, the presentation is confusing. If the top graphs vary the seasonal window with a fixed context window (and the bottom panels do the reverse), why do the fixed window values differ across datasets (e.g., context = 4 for Yahoo vs. context = 48 for NAB)? Could the authors please clarify the experimental setup and the use of different baseline window sizes across datasets?**
>
> A3: Thank you for pointing out this issue. The difference in fixed context window sizes arises from the heterogeneous sampling frequencies across datasets. Specifically:
> - Yahoo: 1 data point per hour
> - WSD / AIOps: 1 data point per minute
> - NAB: 1 data point per 5 minutes
>
> For real-world monitoring data, periodic patterns (e.g., daily or weekly cycles) correspond to real time rather than absolute sequence length. Therefore, datasets with different temporal resolutions naturally contain different numbers of observations per cycle, which directly affects the appropriate choice of window sizes.
>
> Our design principle is to use a context window that captures fine-grained local patterns while ensuring that the total amount of information remains sufficient for decomposition and forecasting. Consequently, the optimal context window differs across datasets due to variations in sampling density and temporal pattern granularity.
>
> Thus, in Section 5.4.2, for each dataset, we select a representative, dataset-appropriate context window, and then vary the seasonal window to study its effect. The fixed context values differ not because of inconsistency, but because each dataset requires a different scale to represent the same real-world temporal granularity. We have clarified this experimental setup in the revised manuscript to avoid potential confusion.
>
> ---
> **Q4: Table 4 reports overall inference time but does not break down the wavelet decomposition overhead. How much time does the noise decomposition step consume relative to the dual-branch prediction?**
>
> A4: Thank you for the question. To clarify the overall pipeline: the wavelet-based noise decomposition module is used only during training, not at inference time.
>
> Concretely, during training, we first apply the noise-decomposition module to the raw input sequence $x$ to obtain a denoised reference sequence $\hat x$. Meanwhile, the raw input $x$ is simultaneously fed into the dual-branch network, which predicts the next timestep’s mean and variance for anomaly scoring. The denoised sequence $\hat x$ is used only when computing the anomaly score or training loss, so that the learned mean and variance are encouraged to reflect the “normal’’ signal (i.e., with noise and outliers suppressed).
>
> At inference time, we feed the raw sequence $x$ directly into the dual-branch network to obtain the predicted normal range (mean and variance) for the next point. The observed value is then compared against this predicted normal range to produce an anomaly decision. Because the noise-decomposition module is not invoked during inference, it introduces zero online latency—this is precisely why our method achieves high inference efficiency.
>
> We measured the time overhead of the noise decomposition step in each epoch during training, as well as its proportion of the total time, and recorded the results in the following table：
>
> | Component            | CPU Time (s) | GPU Time (s) |
> |----------------------|--------------|--------------|
> | Total                | 210          | 39           |
> | Noise Decomposition  | 26.67        | 0.947        |

---

> ### Author Response · Authors · 2025-11-22
> **Response to Reviewer 3pdC (W1-W3)**
>
> **W1: Equation 6 appears to rely on the function or definition presented later in Equation 7. For clarity and logical coherence, the authors should introduce Equation 7 before Equation 6.**
>
> A1: Thanks. We have revised the manuscript accordingly—Equation 7 now precedes Equation 6, and the surrounding text has been polished for clarity.
>
> ---
> **W2: The paper lacks critical implementation details, including specific hyperparameters such as batch size, number of training epochs, window sizes, and decomposition Level L.**
>
> A2: Thank you for highlighting this concern. In the revised manuscript, we have added all relevant hyperparameter settings, including batch size, number of training epochs, window sizes, and the decomposition level L. These parameters are now clearly presented in the Implementation Details section to ensure full transparency and ease of re-implementation.
> | Hyperparameter         | Value |
> |------------------------|-------|
> | batch_size             | 512   |
> | max_epochs             | 30    |
> | seasonal_window_size   | 48    |
> | total_window_size      | 240   |
> | context_window_size    | 4     |
> | d_model                | 256   |
> | num_layers             | 1     |
>
> ---
> **W3: The paper provides a limited explanation for why LSTMs are better than other recent architectures (Transformers, state-space models) that have also demonstrated superior performance on time series tasks.**
> A3: Thank you for raising this important point. We acknowledge that Transformer-based and state-space architectures have shown performance (Lee-Thorp et al., 2022) in multivariate time series (MTS) tasks. For example, models such as Anomaly-Transformer and DCdetector are specifically designed for multivariate anomaly detection, while Informer and Autoformer target multivariate forecasting. These models benefit greatly from the rich feature interactions and high-dimensional structure inherent to MTS data (Zerveas et al., 2021).
>
> However, as discussed in our paper, UTS (univariate time series) presents a fundamentally different setting, where each time step contains only a single scalar value. This leads to：
>
> - Limited information per timestamp
> - Simple and homogeneous distributions
> - Low structural complexity and weaker long-range patterns
>
> These characteristics significantly reduce the expressive advantage of Transformers and state-space models, which typically rely on abundant contextual information to learn meaningful global dependencies. In our experiments (Table 1), we observe that Transformer-based models struggle to establish stable patterns in UTS due to the scarcity of informative features.
> In contrast, LSTMs are naturally suited for UTS because their recurrent state dynamics capture：
>
> - Local dependency, which dominates UTS behavior
> - Smooth temporal transitions, common in low-dimensional signals
> - Short-range pattern memory, which aligns with UTS structural simplicity
>
> Since most UTS patterns arise from local or short-term behaviors, LSTMs provide a better inductive bias (Bai et al., 2018). This also explains why, despite the widespread success of Transformers, LSTMs, and other classical recurrent models remain widely used in low-dimensional time series tasks (Hewamalage et al., 2021).
>
> **Reference**：
> 1. Lee-Thorp J, Ainslie J, Eckstein I, et al. Fnet: Mixing tokens with fourier transforms[C]//Proceedings of the 2022 Conference of the north American chapter of the Association for Computational Linguistics: human language technologies. 2022: 4296-4313.
> 2. Bai S, Kolter J Z, Koltun V. An empirical evaluation of generic convolutional and recurrent networks for sequence modeling[J]. arXiv preprint arXiv:1803.01271, 2018.
> 3. Zerveas G, Jayaraman S, Patel D, et al. A transformer-based framework for multivariate time series representation learning[C]//Proceedings of the 27th ACM SIGKDD conference on knowledge discovery & data mining. 2021: 2114-2124.
> 4. Hewamalage H, Bergmeir C, Bandara K. Recurrent neural networks for time series forecasting: Current status and future directions[J]. International Journal of Forecasting, 2021, 37(1): 388-427.

---

> ### Author Response · Authors · 2025-11-22
> **Response to Reviewer 3pdC (W4-W5)**
>
> **W4: While Section 5.4.1 shows noise decomposition helps, the paper doesn't compare the MAD against other alternatives (pooling-based decomposition, Robust STL or recent decomposition approaches).**
>
> A4: Thank you for this helpful suggestion. We have added the requested ablation experiments in the revised manuscript. Specifically, we now compare our **MAD-based noise decomposition** with several representative alternatives, including **pooling-based decomposition, Robust STL**, and recent decomposition methods.
>
> The **best F1 score** is shown in the table below:
> | Dataset | Noise Decomposition | Pooling Decomposition | STL Decomposition |
> |-|-|-|-|
> | Yahoo | 0.885 | 0.872 | 0.877 |
> | NAB | 0.996 | 0.988 | 0.989 |
> | WSD | 0.910 | 0.865 | 0.874 |
> | AIOPS | 0.936 | 0.923 | 0.914 |
>
>
> The **delay F1 score** is shown in the table below:
> | Dataset | Noise Decomposition | Pooling Decomposition | STL Decomposition |
> |-|-|-|-|
> | Yahoo   | 0.878                | 0.860                  | 0.871             |
> | NAB     | 0.918                | 0.904                  | 0.890             |
> | WSD     | 0.857                | 0.812                  | 0.816             |
> | AIOPS   | 0.879                | 0.871                  | 0.875             |
>
> Train time per epoch:
> | Method               | Time (s) |
> |----------------------|----------|
> | Noise Decomposition  | 52.383   |
> | Pooling Decomposition| 59.332   |
> | STL Decomposition    | 458.389  |
>
> The newly added results show that although all decomposition approaches provide some benefit, **MAD achieves consistently superior performance and better preserves local temporal structure**, validating our design choice.
>
> ---
> **Q5: All datasets come from web/IT monitoring domains, limiting generalizability claims to other time series applications (medical, financial, environmental).**
>
> A5: Thank you for raising this point. We acknowledge the importance of evaluating generalizability across diverse application domains. While our primary focus is the **AIOps (IT operations)** setting—where univariate anomaly detection is a critical real-world challenge, we have additionally evaluated our method on widely used **cross-domain benchmarks**.
>
> Specifically, the **UCR Time Series Classification Archive**, maintained by UC Riverside, is one of the most comprehensive univariate time series collections, covering **medical, environmental, industrial, and financial** domains. Many UCR subsets (e.g., medical ECG, environmental CIMIS44AirTemperature) are standard benchmarks for both classification and anomaly detection. Similarly, the **UCR Anomaly Archive** provides real-world UTS anomaly datasets with precise anomaly segment labels, enabling robust evaluation.
>
> Moreover, the **NAB benchmark** already includes financial indicators and related anomaly patterns. Our experiments demonstrate that **CS-LSTMs** consistently achieve state-of-the-art performance across these datasets.
>
> To further address the reviewer’s concern, we additionally conducted experiments on the **medical (ECG)** and **environmental (CIMIS44AirTemperature)** datasets. As shown in the newly added results, CS-LSTMs continue to outperform mainstream baselines across domains, confirming that the proposed method generalizes well beyond IT monitoring scenarios despite being originally motivated by AIOps challenges.
>
> ECG dataset:
> | Method | Best F1 | Best Precision | Best Recall | Delay F1 | Delay Precision | Delay Recall |
> |-|-|-|-|-|-|-|
> | CS-LSTMs | 0.918 | 0.849 | 1.000 | 0.823 | 0.699 | 1.000 |
> | FCVAE | 0.914 | 0.842 | 0.819 | 0.735 | 0.820 | 0.667 |
> | Anomaly-Transformer | 0.594 | 0.741 | 0.496 | 0.153 | 0.218 | 0.118 |
> | TFAD | 0.835 | 0.783 | 0.894 | 0.713 | 0.684 | 0.745 |
> | Informer | 0.736 | 0.682 | 0.799 | 0.715 | 0.658 | 0.783 |
> | KAN-AD | 0.916 | 0.846 | 1.000 | 0.797 | 0.991 | 0.667 |
> | AnoTransfer | 0.695 | 0.803 | 0.613 | 0.545 | 0.525 | 0.567 |
> | VQRAE | 0.316 | 0.524 | 0.226 | 0.145 | 0.185 | 0.119 |
> | DONUT | 0.513 | 0.420 | 0.391 | 0.445 | 0.405 | 0.494 |
> | SRCNN | 0.740 | 0.705 | 0.779 | 0.636 | 0.602 | 0.674 |
> | SPOT | 0.412 | 0.932 | 0.264 | 0.136 | 0.885 | 0.074 |
>
>
> CIMIS44AirTemperature dataset:
> | Method | Best F1 | Best Precision | Best Recall | Delay F1 | Delay Precision | Delay Recall |
> |-|-|-|-|-|-|-|
> | CS-LSTMs | 0.950 | 0.910 | 0.995 | 0.809 | 0.883 | 0.746 |
> | FCVAE | 0.850 | 0.741 | 0.995 | 0.850 | 0.741 | 0.995 |
> | Anomaly-Transformer | 0.810 | 0.805 | 0.815 | 0.313 | 0.638 | 0.207 |
> | TFAD | 0.772 | 0.718 | 0.835 | 0.756 | 0.733 | 0.780 |
> | Informer | 0.842 | 0.768 | 0.932 | 0.836 | 0.771 | 0.913 |
> | KAN-AD | 0.842 | 0.855 | 0.870 | 0.842 | 0.815 | 0.870 |
> | AnoTransfer | 0.562 | 0.752 | 0.449 | 0.540 | 0.608 | 0.486 |
> | VQRAE | 0.504 | 0.688 | 0.398 | 0.494 | 0.672 | 0.391 |
> | DONUT | 0.255 | 0.410 | 0.185 | 0.255 | 0.410 | 0.185 |
> | SRCNN | 0.285 | 0.310 | 0.264 | 0.212 | 0.246 | 0.186 |
> | SPOT | 0.432 | 0.560 | 0.352 | 0.448 | 0.542 | 0.382 |

---

> ### Author Response · Authors · 2025-11-26
> **Response to Reviewer 3pdC**
>
> Dear Reviewer,
>
> We'd appreciate it if you'd let us know if our response has addressed your concerns.
>
> Thanks!

---

### Author Response · Authors · 2025-12-01
**Responses Overview**

Dear Area Chair and Reviewers,

We sincerely thank all reviewers for their constructive comments. We have revised the manuscript to clarify methodology, improve structure, and include new experimental results. Below, we summarize the main clarifications and improvements.

---

## 1. Clarifications to Reviewers’ Questions

### (1) Noise Decomposition & Masked Loss Pipeline
- Added a full mathematical description of the wavelet-based noise decomposition (MAD thresholding).
- Clarified that decomposition is **only used during training**, thus introducing **zero inference overhead**.
- Provided complete mask construction procedures for both **supervised** and **unsupervised** settings.

### (2) Dual-Branch Architecture (S-LSTM & C-LSTM)
- Added detailed data flow diagrams and formulas.
- Clarified how time-domain and frequency-domain features are fed into both branches.
- Formally defined the *evolution of periodicity* using frequency-domain stability and LSTM dynamics.

### (3) Window Sizes, Hyperparameters, and Dataset Differences
- Explained dataset-specific window choices due to different sampling intervals (1 min / 5 min / 1 hr).
- Provided tuning guidelines and demonstrated robustness to window-size variation.

### (4) Decomposition Level \(L\)
- Derived $L = \lfloor \log_2(N) \rfloor$ using wavelet decomposition theory and cited classical references.

### (5) LSTM vs. Transformer / State-Space Models
- Explained why Transformers and SSMs underperform on **univariate** time series (low information density, weak long-range structure).
- Added theoretical and empirical justification for LSTM inductive-bias advantages in UTS settings.

### (6) Comparison with TFAD / Dual-TF / WaveletAE
- Conducted both qualitative and quantitative comparisons of the four methods, analyzed the differences in their architectural designs, and presented their accuracy and efficiency in a structured table.
- Highlighted that CS-LSTMs are the first to **explicitly model the temporal evolution of periodicity**.

### (7) Generalization to Diverse Anomaly Types
- Clarified that benchmark datasets naturally include:
  - regime shifts
  - variance changes
  - spikes
  - contextual anomalies
  - non-seasonal trends
- Explained why the dual-branch predictive design generalizes well to these diverse anomaly forms.

---

## 2. Improvements in Manuscript

### (1) Unified Notation Throughout the Paper
- Replaced ambiguous Kronecker notation with correct element-wise multiplication.
- Reordered Equation (6) and Equation (7) for logical consistency.

### (2) Comprehensive Hyperparameter Table Added in Appendix A

### (3) New Ablations on the Denoising Strategy (Section 5.4.2)
- Added comparisons with pooling-based decomposition, STL decomposition, and recent methods.
- Results show **MAD decomposition consistently achieves the best F1 and Delay-F1**, while being substantially faster.

### (4) Training & Inference Efficiency Experiments (Section 5.2, Appendix G)
- Added absolute latency and memory benchmarks (CPU/GPU).
- CS-LSTMs exhibit the **fastest inference**, and training is competitive despite decomposition overhead.

### (5) Cross-Domain Experiments Added in Appendix H
- Included full evaluations on:
  - **ECG (medical)**
  - **CIMIS44 Air Temperature (environmental)**
- CS-LSTMs outperform baselines across domains, confirming strong generalizability.

---

We sincerely appreciate the reviewers’ input, which has significantly strengthened the manuscript, and we hope our revisions satisfactorily address all concerns.

---

### Meta-Review · Area_Chair_icab · 2025-12-29

**Summary:**

This submission proposes CS-LSTMs, a dual-branch LSTM framework for univariate time-series AD that separates frequency- and time-domain dynamics and incorporates a wavelet-based noise decomposition used during training.

Reviewers find the problem well motivated, the architecture clear, and the empirical results strong (3pdC, S6ae, Bshn). In the rebuttal, the authors substantially clarified the methodology, added missing implementation details and ablations, reported absolute efficiency metrics, and provided quantitative comparisons to recent time-frequency baselines raised in discussion (3pdC; S6ae). The remaining concerns mainly relate to the degree of architectural novelty and the univariate scope (Bshn, low confidence; partially S6ae), but these do not outweigh the paper’s technical soundness and empirical strength, leading to a recommendation to accept.


Comment to the authors:

Some dataset references are underspecified or not currently verifiable (e.g., “Wsd dataset” and “Yahoo dataset”). Please revise these citations to include clear dataset names/identifiers, appropriate attribution, and stable, reachable links.

**Reviewer Concerns:**

3pdC: Clarity issues (e.g., equation ordering, lack of mathematical descriptions). Addressed

3pdC: Reproducibility gaps. Addressed

3pdC, S6ae: Insufficient model justification. Addressed

3pdC: Limited dataset diversity and generalizability. Addressed

S6ae, Bshn: Missing analysis of scalability to multivariate scenarios. Partly Addressed

S6ae: Computational efficiency reporting. Addressed

S6ae: Missing recent baselines comparison. Addressed

Bshn: Generalization beyond the targeted anomaly types. Addressed

Bshn: Hyperparameter sensitivity & robustness. Addressed

**Reviewer Scores:**

3pdC: Probably would have raised from 4 to 6

S6ae: Probably would have stayed at 6

Bshn: Would remain at 4 (explicitly maintains original score).

---

### Decision · Program_Chairs · 2026-01-26

Accept (Poster)